# GPSToken: Gaussian Parameterized Spatially-adaptive Tokenization for Image Representation and Generation

**Zhengqiang Zhang**[1,2]**, Rongyuan Wu**[1,2]**, Lingchen Sun**[1,2]**, Lei Zhang**[1,2,†]

[1]The Hong Kong Polytechnic University     [2]OPPO Research Institute

`zhengqiang.zhang,rong-yuan.wu,ling-chen.sun@connect.polyu.hk,`
`cslzhang@comp.polyu.edu.hk`

[†]Corresponding author

https://github.com/xtudbxk/GPSToken

## Abstract

Effective and efficient tokenization plays an important role in image representation and generation. Conventional methods, constrained by uniform 2D/1D grid tokenization, are inflexible to represent regions with varying shapes and textures and at different locations, limiting their efficacy of feature representation. In this work, we propose **GPSToken**, a novel **G**aussian **P**arameterized **S**patially-adaptive **Token**ization framework, to achieve non-uniform image tokenization by leveraging parametric 2D Gaussians to dynamically model the shape, position, and textures of different image regions. We first employ an entropy-driven algorithm to partition the image into texture-homogeneous regions of variable sizes. Then, we parameterize each region as a 2D Gaussian (mean for position, covariance for shape) coupled with texture features. A specialized transformer is trained to optimize the Gaussian parameters, enabling continuous adaptation of position/shape and content-aware feature extraction. During decoding, Gaussian parameterized tokens are reconstructed into 2D feature maps through a differentiable splatting-based renderer, bridging our adaptive tokenization with standard decoders for end-to-end training. GPSToken disentangles spatial layout (Gaussian parameters) from texture features to enable efficient two-stage generation: structural layout synthesis using lightweight networks, followed by structure-conditioned texture generation. Experiments demonstrate the state-of-the-art performance of GPSToken, which achieves rFID and FID scores of 0.65 and 1.50 on image reconstruction and generation tasks using 128 tokens, respectively. Codes and models of GPSToken can be found at https://github.com/xtudbxk/GPSToken.

## 1 Introduction

Recent advances in latent generative models such as VQGAN [11], LDM [29], MaskGIT [3], DiT [26], SiT [23], VAR [32], and SD3 [10] have revolutionized the research and application of image generation. Most of these methods adopt a two-stage framework. First, an auto-encoder is employed to convert original images into compact latent representations with reduced dimensionality (*e.g.*, 256×256 → 32×32 in LDM), serving as an effective "image tokenizer". Then, generative models [11, 29, 3, 26, 23, 32, 10, 40] are trained in the latent space, alleviating computational burdens while enabling high-quality generation. The primary goal of an image tokenizer is to learn effective representations through reconstruction tasks, encoding images into a latent space with minimal loss. Early methods such as VAE [20] transform images into continuous latent spaces. LDM [29] performs diffusion in the latent space, reducing computational cost while improving visual

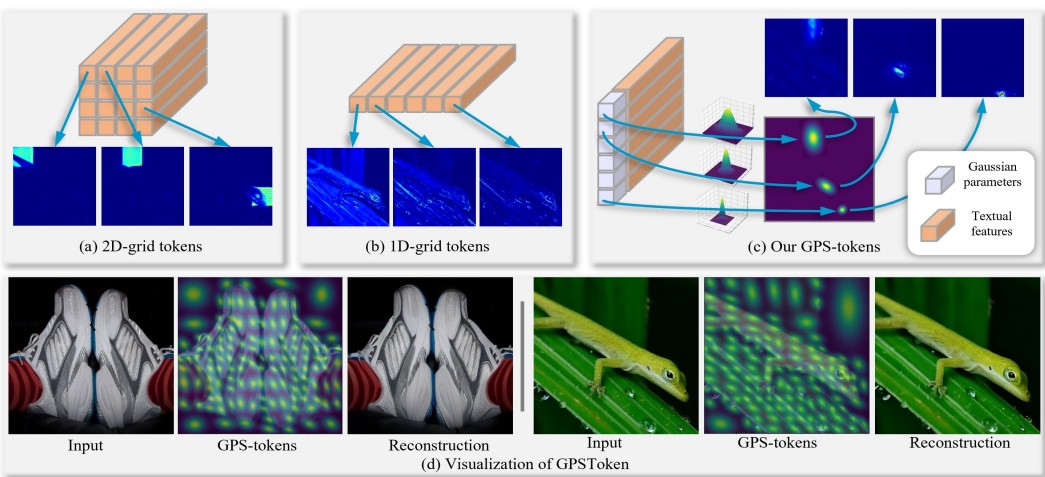

Figure 1: Comparisons between (a) 2D-grid tokens, (b) 1D-grid tokens and (c) our GPS-tokens. (d) Two visualization examples of the representation and reconstruction results of GPSToken.

quality. In contrast to continuous representation, VQVAE [33] introduces discrete latent codes via vector quantization. Based on VQVAE, VQGAN [11] and MaskGiT [3] train autoregressive models and achieve improved image generation performance. Beyond 2D grid tokenization, TiTok [37] transforms images into compact 1D latent sequences, significantly reducing tokens. FlexTok [1] and One-D-Piece [24] dynamically adjust counts, enhancing efficiency by concentrating key information early in the sequence. MAETok [5] shows masked autoencoders yield discriminative latent spaces suitable for diffusion models.

Despite the significant progress achieved by existing image tokenization methods, the grid-based tokenization strategy used by them is inefficient and inflexible in representing the different regions with different contents in natural images. As illustrated in Figs. 1 (a) and (b), 2D-grid tokens represent local patches of fixed size and at fixed positions, no matter whether the patch has complex structures or details, while 1D-grid tokens encode globally contextualized information from the entire image, lacking spatially-adaptive representation ability.

In this paper, we propose **GPSToken**, a novel **G**aussian **P**arameterized **S**patially-adaptive **To-ken**ization framework, to achieve non-uniform and flexible image tokenization. GPSToken parameterizes each token with a 2D Gaussian function, encoding both the positions and shapes of different regions in an image. Specifically, as shown in Fig. 1 (c), each GPS-token consists of two components: the first component stores the standard deviation and position of the Gaussian function, representing region shape and location, while the second component represents the textural features of the region. Inspired by Gaussian Splatting [18], our GPSToken can be rendered into 2D feature maps, facilitating seamless integration with conventional 2D decoders and enabling end-to-end training.

To achieve spatially-adaptive tokenization, we iteratively partition an image into regions of varying sizes and shapes. The partitioned regions, though having different shapes and positions, will have a similar amount of information in terms of entropy. The shape and position of each region are used to initialize the corresponding Gaussian parameters. Then, we utilize a transformer, for which each query corresponds to a token, to refine these parameters and extract textural features. The shape-texture decomposition of GPSToken provides distinct advantages for image generation. With GPSToken, we can first generate Gaussian parameters that encode spatial layout (region shape/position), then synthesize texture features conditioned on the Gaussian geometric priors. The Gaussian priors act as structural constraints, simplifying texture generation while ensuring spatial consistency. This shape-texture decomposition approach aligns with how humans conceptualize images (structure-first, details-later), accelerating the model training and improving the generation quality.

Our method achieves significant improvements over existing methods in both image reconstruction and generation tasks. For image reconstruction, GPSToken achieves "rec. FID", PSNR and SSIM scores of 0.65, 24.06 and 0.657 on the ImageNet 256×256 reconstruction task using 128 tokens. For image generation, our model achieves a state-of-the-art FID of 1.50 on the ImageNet 256 generation task, surpassing recent methods such as Titok [37], FlexTok [1], One-D-Piece [24] and MAETok [5]. Our contributions are summarized as follows:

- We propose GPSToken, an effective Gaussian parameterized spatially-adaptive tokenization method for image representation and generation. GPSToken leverages 2D Gaussian functions to dynamically model varying region shapes and positions, significantly reducing representation redundancy in simple regions while achieving finer representation in texture-rich regions.

- With GPSToken, we present a shape-texture decomposition method for image generation, reducing generation complexity, accelerating model training, and improving generation quality.

- Extensive experiments validate the effectiveness of GPSToken. Our work paves the way toward effective and efficient spatially-adaptive image representations, benefiting a variety of vision tasks.

## 2   Related Work

**Latent Generative Models**. Latent models have gained significant attention in visual generation. VAE [20] constructs continuous latent spaces with Gaussian priors, while VQVAE [33] couples codebooks with autoregressive modeling for discrete latent representation. VQGAN [11] incorporates adversarial training and transformer-based autoregressive components, further improving generative performance. MaskGiT [3] refines discrete latent generation through scheduled parallel sampling, significantly accelerating inference. LDM [29] enables high-resolution synthesis by embedding diffusion in compressed latent spaces. DiT [26] demonstrates transformer scalability in latent diffusion, and SiT [23] extends DiT with flexible interpolation, offering versatile distribution mapping.

**Image Tokenization**. Image tokenization aims to create compact representations of high-dimensional images. Early methods often use VAE [20] for continuous tokenization and VQVAE [33] for discrete tokenization. VQVAE-2 [28] introduces a multi-scale structure, while RQVAE [21] builds extra codebooks to quantize residuals. DCAE [7] ensures quality at high compression ratios. MaskBit [34] proposes an embedding-free autoencoder using bit tokens. Recently, 1D grid-based tokenization has gained attention for more compact representations. TiTok [37] is among the first to convert 2D images into 1D latent tokens using masked transformers for encoding and decoding. SoftVQ [6] uses soft categorical posteriors to combine multiple codewords into one continuous token. FlexTok [1] and One-D-Piece [24] project 2D images into variable-length, ordered 1D sequences, allowing good reconstructions. MaeTok [5] leverages mask modeling to learn semantically rich and reconstructive latent spaces, highlighting the importance of space structure for generation.

Despite the significant progress, grid-based methods remain inefficient and inflexible in capturing regions with varying content. To address this, we propose GPSToken, which parameterizes each token using a 2D Gaussian function to encode region positions and shapes, allowing spatially adaptive alignment with local texture complexity. Note that while GaussianToken [9] also uses 2D Gaussians, it simply replaces the original tokens in VQVAE with Gaussian distributions without spatial adaptivity. Besides, our GPSToken can decouple the visual generation process into layout synthesis and texture feature generation, while GaussianToken does not possess a corresponding generator.

## 3   Methodology

In this section, we first describe the parameterization of GPSToken using 2D Gaussian functions, then present the detailed training procedure for obtaining GPSToken. The resulting tokens can be transformed into pixel-domain images through a decoder. Finally, leveraging the inherent shape-texture decomposition property of GPSToken, we propose a two-stage image generation pipeline to accelerate the training of generative models while improving their performance.

### 3.1   Gaussian Parameterized Tokenization

Processing images in pixel space is computationally expensive and increases model complexity. To reduce cost, existing methods [33, 28, 37, 5] employ image tokenizers that project an image $\mathbf{x} \in \mathbb{R}^{H \times W \times 3}$ into low-dimensional tokens $\mathbf{z} \in \mathbb{R}^{l \times c}$, with $l \ll H \times W$. However, current 2D/1D tokenizers are limited by rigid grid structures, restricting flexible representation of regions with varying sizes and contents. We propose **GPSToken**, a novel method that parameterizes tokens using 2D Gaussian functions, enabling adaptive and efficient modeling of complex visual regions.

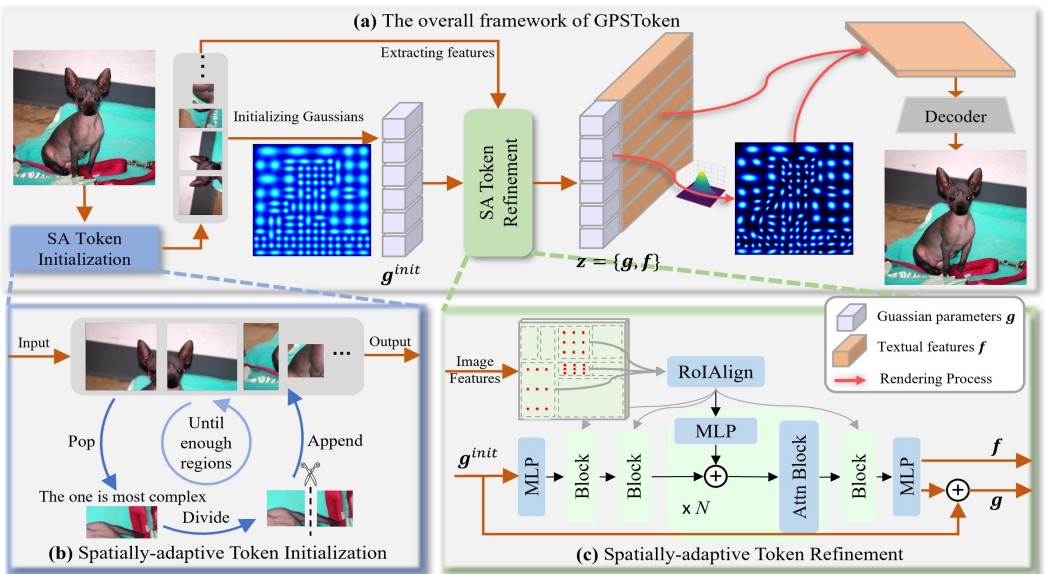

Figure 2: (a) The overall framework of our GPSToken. (b) Spatially-adaptive Token Initialization. (c) Spatially-adaptive Token Refinement.

**2D Gaussian Parameterized Tokens.** A standard 2D Gaussian function $p(x, y)$ is given by:

$$p(x,y) = \frac{\hat{p}(x,y)}{Z} = \frac{1}{Z} \exp\left(-\frac{1}{2(1-\rho^2)}\left(\frac{(x-\mu_x)^2}{\sigma_x^2} - \frac{2\rho(x-\mu_x)(y-\mu_y)}{\sigma_x\sigma_y} + \frac{(y-\mu_y)^2}{\sigma_y^2}\right)\right), \quad (1)$$

where $Z$ is the normalization constant, $\sigma_x, \sigma_y > 0$ are the standard deviations along the $x$- and $y$-axes, and $\rho \in [-1, 1]$ denotes the correlation coefficient. The means $\mu_x, \mu_y \in \mathbb{R}$ determine the center.

To reduce computation and focus on local regions, we modify the standard 2D Gaussian by restricting its spatial support to a bounded region centered at $(\mu_x, \mu_y)$ and omitting the normalization constant. This design removes unnecessary computation while preserving fine details in the region of interest. Specifically, the modified Gaussian function is defined as:

$$\mathbf{g}(x,y) = \begin{cases} \hat{p}(x,y), & \text{if } |x-\mu_x| \leq s\sigma_x \text{ and } |y-\mu_y| \leq s\sigma_y, \\ 0, & \text{otherwise}, \end{cases} \quad (2)$$

where $s$ is a hyperparameter controlling the spatial support of the Gaussian function.

Using $\mathbf{g}$, we represent an image $\mathbf{x}$ via $l$ Gaussian parameterized tokens $\mathbf{z} \in \mathbb{R}^{l \times c}$, as shown in Fig. 1 (c). Each token contains two components $\mathbf{z}_i = \{\mathbf{g}_i, \mathbf{f}_i\}$. The first component $\mathbf{g}_i = \{\sigma_x^{(i)}, \sigma_y^{(i)}, \rho^{(i)}, \mu_x^{(i)}, \mu_y^{(i)}\}$ (gray cuboids in Fig. 1 (c)) encodes spatial position and deviation of the Gaussian function. The second component $\mathbf{f}_i \in \mathbb{R}^{(c-5)}$ (orange cuboids) holds texture features, capturing detailed visual information from the corresponding region. This enables joint encoding of geometric and visual characteristics across image regions.

**Splatting-Based Rendering.** Inspired by GS [18, 4], we render GPS-tokens into 2D feature maps using splatting-based rendering. This is possible because each 2D Gaussian is continuous and can be sampled into 2D features. For example, given $l$ Gaussian-parameterized tokens $\{\mathbf{z}_0, \mathbf{z}_1, \cdots, \mathbf{z}_{l-1}\}$, the $k$-th channel of the rendered 2D feature map at $(x, y)$ can be obtained as follows:

$$R(x,y,k) = \sum_{i=0}^{l-1} r_i(x,y,k) = \sum_{i=0}^{l-1} \mathbf{g}_i(x,y) \times \mathbf{f}_i[k]. \quad (3)$$

**Advantages over Bounding Boxes and Segmentation Maps.** Alternative approaches to representing image regions often rely on bounding boxes or segmentation maps. Bounding boxes define regions using axis-aligned rectangles, while segmentation maps assign discrete labels to individual pixels. Compared with them, our Gaussian-parameterized tokenization offers several key advantages. First, each 2D Gaussian models anisotropic shapes with only five parameters $(\mu_x, \mu_y, \rho, \sigma_x, \sigma_y)$, enabling a compact and geometry-adaptive representation that is both expressive and lightweight – reducing

the burden on downstream tasks. Second, the Gaussian function provides a smooth, continuous weight distribution over pixels, naturally capturing uncertainty and modeling soft or ambiguous boundaries in natural images. Third, GPSToken is fully differentiable, enabling end-to-end training and seamless integration into existing gradient-based learning frameworks. In contrast, bounding boxes are restricted to rigid, axis-aligned shapes and exhibit hard, non-differentiable boundaries. Segmentation maps, while precise, are high-dimensional, discrete, and inherently incompatible with differentiable optimization.

## 3.2 Spatially-adaptive GPSToken Learning

Image tokenizers typically use an encoder-decoder framework, where the encoder maps the image $\mathbf{x}$ to a latent representation $\mathbf{z} = \text{Enc}(\mathbf{x})$, and the decoder reconstructs it as $\hat{\mathbf{x}} = \text{Dec}(\mathbf{z})$. Our GPSToken also follows this framework. As shown in Fig. 2 (a), we first apply an iterative algorithm to partition the image into regions of varying sizes based on texture complexity. Each region's position and size initialize the Gaussian parameters of the corresponding GPS-tokens, providing a coarse spatially-adaptive representation. Next, a transformer-based encoder refines GPS-tokens for fine-grained adaptation, adjusting the position, shape, and orientation according to regional textures. Finally, the GPSTokens are converted back to 2D feature maps and passed through a decoder to reconstruct $\hat{\mathbf{x}}$.

**Spatially-adaptive Token Initialization.** As shown in Fig. 2 (b), we use an iterative algorithm to initialize Gaussian parameters aligned with local regions. Specifically, we maintain a dynamic list of region candidates and iteratively split the most complex regions into simpler sub-regions until the target number is reached. We measure region complexity using gradient entropy. We compute the gradient magnitude map $E$ via the Sobel operator [31], then calculate the information entropy $H$ from the histogram of $E$. The overall metric is defined as:

$$m = hw \times H^\lambda = hw \times \left( -\sum_{i=1}^{512} q_i \log(q_i) \right)^\lambda, \tag{4}$$

where $h$ and $w$ are the spatial size of regions, $q_i$ is the probability of gradients in the $i$-th histogram bin, and $\lambda$ balances size and complexity. By integrating region size into the metric, we promote division of larger regions. A higher $m$ value indicates a larger and more complex region.

Once regions are determined, we associate the $i$-th GPSToken $\mathbf{z}_i$ with the $i$-th region and initialize its Gaussian parameters as $\mathbf{g}_i^{init} = \{\sigma_x^{(i)}, \sigma_y^{(i)}, \rho^{(i)}, \mu_x^{(i)}, \mu_y^{(i)}\} = \{\frac{w_i}{6}, \frac{h_i}{6}, 0, x_i, y_i\}$, where $h_i$, $w_i$ are the height and width of regions, and $(x_i, y_i)$ is its center. Setting $\sigma_x^{(i)}$ and $\sigma_y^{(i)}$ to $\frac{1}{6}$ of $w_i$ and $h_i$ ensures full coverage during rendering. Please see Algorithm 1 in the **Appendix** for more details.

**Spatially-adaptive Token Refinement.** After obtaining the initialized Gaussian parameters, we employ a transformer-based encoder to refine these parameters to achieve fine-grained spatial adaptation, while simultaneously extracting the corresponding texture features $\mathbf{f}$ for each region.

As shown in Fig. 2(c), the encoder first projects the initial Gaussian parameters $\mathbf{g}^{init}$ into query embeddings, which are then processed by attention blocks. To focus each embedding on its corresponding region, we include region-specific features as conditions. Specifically, we extract image features via residual blocks and use RoIAlign [12] to obtain fixed-size features for each region. These are added to the query embeddings before each attention block. This ensures that each query interacts with its local image features, improving alignment with regional textures.

Additionally, self-attention blocks enable query embeddings to interact with each other, considering the global image layout during training. The encoder outputs residuals $\Delta\mathbf{g}$ for refining Gaussian parameters and textual features $\mathbf{f}$ for each token. The final GPS-tokens are:

$$\mathbf{z} = \{\mathbf{g}^{init} + \Delta\mathbf{g}, \mathbf{f}\}. \tag{5}$$

The refined Gaussian parameters $\mathbf{g}$ define the spatial layout and overall structure of the image, while $\mathbf{f}$ encode the textual patterns of Gaussians. They work synergistically to represent the whole image.

To illustrate the spatial adaptation of GPSTokens, we visualize $\mathbf{g}^{init}$ and $\mathbf{g}$ as Gaussian maps in Fig. 2. As shown in Fig. 2(a), the initial map $\mathbf{g}^{init}$ aligns with the region partitions. Complex regions have denser Gaussians, while simpler ones use fewer, larger Gaussians. After encoder refinement, the parameters better match local textures. While $\mathbf{g}^{init}$ contains only axis-aligned Gaussians, the refined $\mathbf{g}$ includes rotated ones that align better with local structures, such as the dog's ear edges.

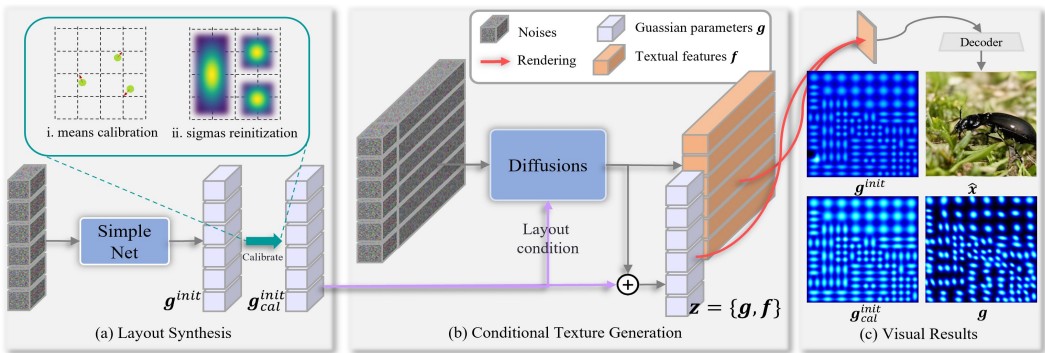

Figure 3: The overview of two-stage generation pipeline based on GPSToken.

During decoding, we first render the GPSTokens $\mathbf{z}$ into a 2D feature map using Eq. 3, then decode them into the reconstructed image. Following VQGAN [11], we use a combination of reconstruction loss $L_{\text{rec}}$, perceptual loss $L_{\text{perc}}$, and adversarial loss $L_{\text{adv}}$ during training.

### 3.3 GPSToken-driven Two-stage Image Generation

The shape-texture decomposition property of GPSToken naturally offers a two-stage image generation pipeline, which first synthesizes the image layout using the shape information and then generates the image details using the texture features. This two-stage scheme simplifies the image generation process and improves the generation quality.

**Layout Synthesis.** As illustrated in Fig. 3 (a), in the first stage, we focus on generating the overall structure of the image, which can be represented by the Gaussian parameters. Note that we use the initial Gaussian parameters $\mathbf{g}^{init}$, instead of the final parameters $\mathbf{g}$, for layout synthesis. This is because the initial Gaussian parameters are more decorrelated with the local textures and are easier to predict, while they are good enough to represent the image rough layout. Specifically, we first generate the $\mathbf{g}^{init}$ using a simple generative model and then calibrate it to correct potential inaccuracies (see Fig. 3 (c)). The calibration procedure consists of two steps: calibrate the means $\{\mu_x, \mu_y\}$ of each Gaussian to its nearest valid values and recompute $\{\sigma_x, \sigma_y, \rho\}$, obtaining the calibrated Gaussian parameters $\mathbf{g}^{init}_{cal}$. The detailed calibration can be found in Algorithm 2 in the **Appendix**.

**Texture Generation.** After synthesizing the overall structure of the image, we enrich the generated layout with detailed textures (see Fig. 3 (b)) using diffusion models such as SiT [23]. Specifically, we first convert $\mathbf{g}^{init}_{cal}$ into embedding vectors and incorporate them as additional inputs in each timestep. This ensures that the newly generated texture features accurately reflect the constraints of the original layout while preserving structural consistency and rich details in the final image. In practice, the model predicts a Gaussian parameter residual $\Delta\mathbf{g}$ and texture features $\mathbf{f}$. The Gaussian parameters are updated by $\mathbf{g} = \mathbf{g}^{init}_{cal} + \Delta\mathbf{g}$, while $\mathbf{f}$ captures the specific texture characteristics of each token. The results can be rendered and reconstructed into natural images using our GPSToken decoder.

The two-stage generation pipeline significantly reduces the complexity of image generation by decoupling geometric modeling from texture synthesis. The $\mathbf{g}^{init}_{cal}$ acts as a structural constraint, simplifying the texture generation task while ensuring spatial consistency. It aligns with the human perception process (from structure to detail). Additionally, since layout synthesis is much easier than texture generation, we employ a simple network or cached database to produce results in the first stage, introducing minimal additional computation compared to existing methods.

## 4 Experiments

### 4.1 Experimental Settings

**Training Data and Settings.** We train all models on the ImageNet dataset [30], which contains 1.28M training images and 50K validation images. During preprocessing, images are resized to $256 \times 256$ and center-cropped without additional augmentation beyond horizontal flipping. We implement three variants of GPSToken: GPSToken-S64 (64 tokens), GPSToken-M128 (standard setting, used

Table 1: Comparisons of $256 \times 256$ reconstruction task on Imagenet val set. The top 3 methods *trained only with ImageNet* are highlighted in red, blue and green. Note that "SDXL-VAE" is trained with a rich amount of additional data other than Imagenet.

| Method | Tokens | Params (M) | sample-level | | | distribution-level | | | |
|---|---|---|---|---|---|---|---|---|---|
| | | | PSNR ↑ | SSIM ↑ | LPIPS ↓ | rec. FID ↓ | rec. sFID ↓ | FID ↓ | sFID ↓ |
| **2D Tokenization** | | | | | | | | | |
| SDXL-VAE [27] | 32×32 | 83.6 | 25.55 | 0.727 | 0.066 | 0.73 | 2.42 | 2.35 | 3.89 |
| GaussianToken [18] | 32×32 | 130.6 | 22.40 | 0.597 | 0.112 | 1.70 | 4.62 | 3.63 | 4.71 |
| VQVAE-f16 [11] | 16×16 | 89.6 | 19.41 | 0.476 | 0.191 | 8.01 | 9.64 | 10.74 | 7.38 |
| MaskGIT-VAE [3] | 16×16 | 54.5 | 18.11 | 0.427 | 0.202 | 3.79 | 5.81 | 5.19 | 4.56 |
| VAVAE [35] | 16×16 | 69.8 | 25.76 | 0.742 | 0.050 | 0.27 | 1.72 | 1.74 | 3.91 |
| DCAE [7] | 8×8 | 323.4 | 23.62 | 0.644 | 0.092 | 0.98 | 4.82 | 2.59 | 5.02 |
| **1D Tokenization** | | | | | | | | | |
| SoftVQ [6] | 64 | 173.6 | 21.93 | 0.568 | 0.115 | 0.92 | 4.52 | 2.51 | 4.21 |
| TiTok-B64 [37] | 64 | 204.8 | 17.01 | 0.390 | 0.263 | 1.75 | 4.51 | 2.50 | 4.21 |
| TiTok-S128 [37] | 128 | 83.7 | 17.66 | 0.413 | 0.220 | 1.73 | 7.25 | 3.25 | 5.52 |
| MAETok [5] | 128 | 173.9 | 23.25 | 0.626 | 0.096 | 0.65 | 3.87 | 2.01 | 4.39 |
| FlexTok [1] | 256 | 949.7 | 17.69 | 0.475 | 0.257 | 4.02 | 8.00 | 4.88 | 6.12 |
| One-D-Piece [24] | 256 | 83.9 | 17.74 | 0.420 | 0.210 | 1.54 | 6.96 | 2.93 | 5.36 |
| MaskBit [34] | 256 | 54.5 | 21.07 | 0.539 | 0.142 | 1.29 | 4.72 | 3.08 | 4.09 |
| **GPSToken** | | | | | | | | | |
| **S64** | 64 | 127.5 | 22.18 | 0.578 | 0.111 | 1.31 | 5.42 | 3.02 | 4.85 |
| **M128** | 128 | 127.8 | 24.06 | 0.657 | 0.080 | 0.65 | 3.28 | 2.18 | 3.96 |
| **L256** | 256 | 128.7 | 28.81 | 0.809 | 0.043 | 0.22 | 1.31 | 1.65 | 3.77 |

as default), and GPSToken-L256 (256 tokens). For more details about the training/inference and network architectures, please refer to **Appendix**.

**Evaluation Metrics.** We conduct all evaluations on the validation set of ImageNet. For *reconstruction*, we evaluate performance using both **sample-level** and **distribution-level** indices. At the sample level, we use PSNR, SSIM, and LPIPS [39], which measure the similarity between reconstructed and original images, as metrics. At the distribution level, we adopt FID [13] and sFID [25] to assess the overall distribution of reconstructed images. Specifically, we report "rec. FID" and "rec. sFID" to measure the distribution consistency between reconstructed and input images, while using standard "FID" and "sFID" to evaluate the alignment with natural image distributions. For image *generation*, we employ FID to assess generation quality.

## 4.2 Image Representation

**Comparison Results.** We evaluate the representation performance of GPSToken using the image reconstruction task. We compare GPSToken with existing 1D and 2D tokenization methods at $256 \times 256$ resolution, including SDXL-VAE [11], GaussianToken [9], VQVAE [11], MaskGiT-VAE [3], VAVAE [35], DCAE [7], TiToK [37], SoftVQ [6], FlexTok [1], One-D-Piece [24], and MAETok [5]. As shown in Table 1, GPSToken-L256 achieves significantly better performance than the competing methods across both sample-level and distribution-level metrics, even better than SDXL-VAE, which utilizes more tokens (1024 vs. 256) and is trained with a rich amount of additional data. Compared to SDXL-VAE, GPSToken-L256 improves PSNR by 3.26, SSIM by 0.082, and reduces LPIPS by 0.023. It also achieves a "rec. FID" of 0.22, a "rec. sFID" of 1.31, an FID of 1.65, and an sFID of 3.77, outperforming all competitors. Note that VAVAE [35] leverages vision foundation models to align latent features, yet it still lags behind GPSToken-L256.

On the other hand, GPSToken-M128 outperforms the competing methods using the same number of tokens on most metrics, obtaining a "rec. sFID" of 3.28 and LPIPS of 0.080. It also outperforms many methods that use more tokens. With only 64 tokens, GPSToken-S64 also demonstrates promising performance, achieving a "rec. FID" score of 1.31, highlighting the scalability of our approach.

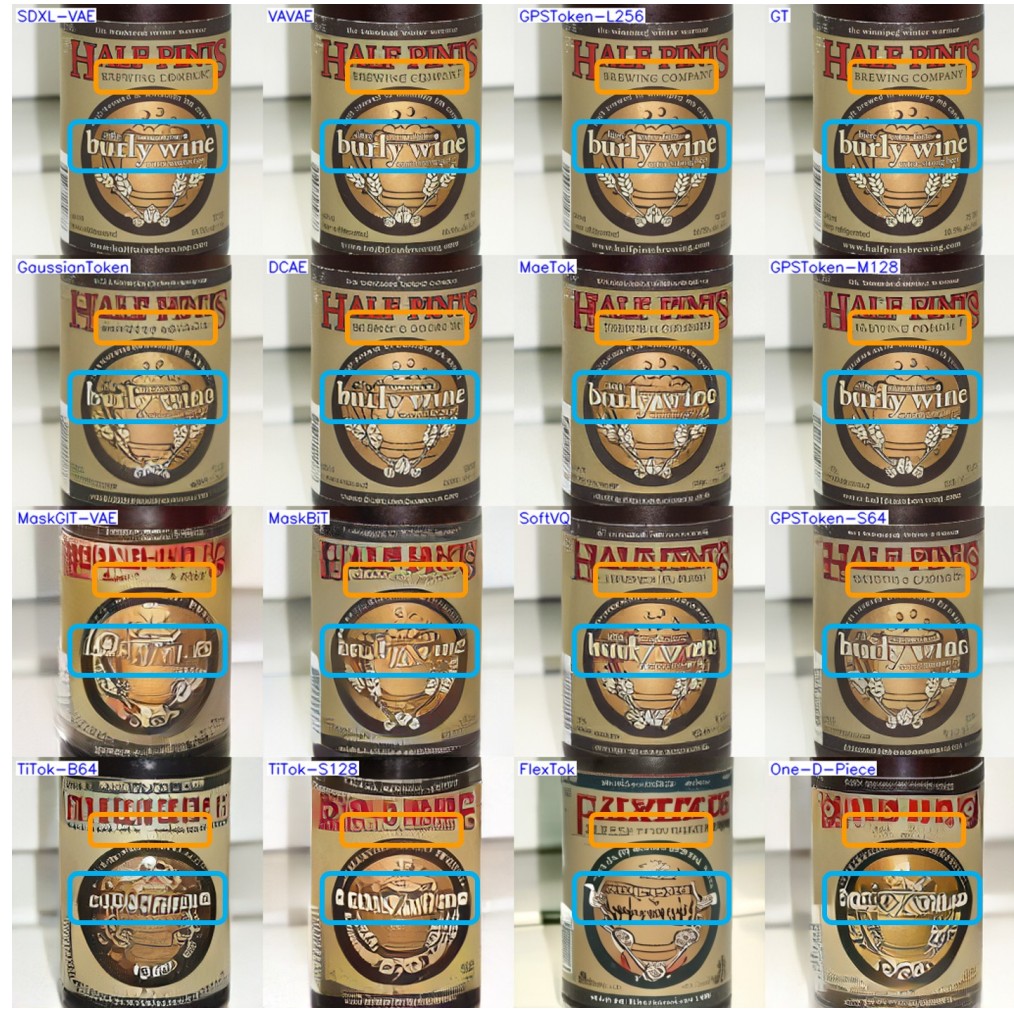

Figure 4: **Visual comparisons on** $256 \times 256$ **reconstruction task.**

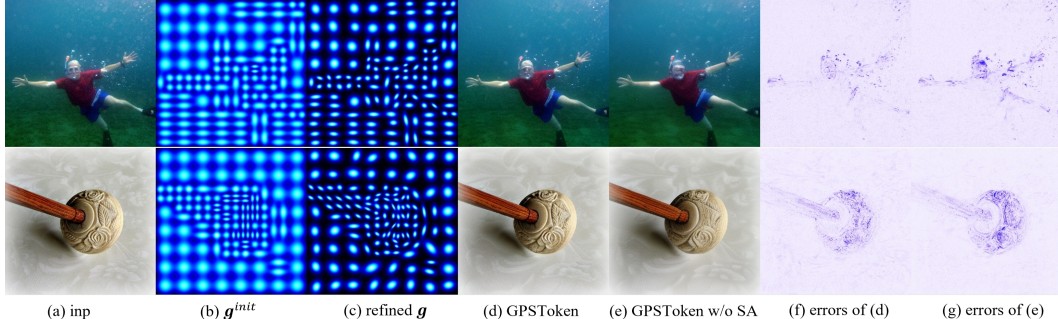

| (a) inp | (b) $\boldsymbol{g}^{init}$ | (c) refined $\boldsymbol{g}$ | (d) GPSToken | (e) GPSToken w/o SA | (f) errors of (d) | (g) errors of (e) |

Figure 5: **Illustration of Spatial Adaptivity (SA)**. Left to right: the input $\mathbf{x}$, visualization of $\mathbf{g}^{init}$, visualization of refined $\mathbf{g}$, the reconstruction $\hat{\mathbf{x}}$ with SA, the reconstruction $\hat{\mathbf{x}}_{\text{w/o SA}}$ without SA, error map of $\hat{\mathbf{x}}$, and error map of $\hat{\mathbf{x}}_{\text{w/o SA}}$ (darker blue indicates larger errors).

We provide visual comparisons among GPSToken and its competitors in Figs. 4. It can be observed that our GPSToken achieves significantly more accurate and clearer textures in complex regions, without compromising the performance in simpler areas.

**Effectiveness of Spatial Adaptivity.** GPSToken possesses spatial adaptivity (SA), enabling a region adaptive image representation. As shown in Fig. 5, with our SA initialization, the $\mathbf{g}^{init}$ is placed according to the regional complexity. More Gaussians are assigned to complex regions such as the human body, while sparse Gaussians are used in simpler regions. Based on this initialization, the refined $\mathbf{g}$ further adjusts its positions, shapes, and orientations to better align with local textures.

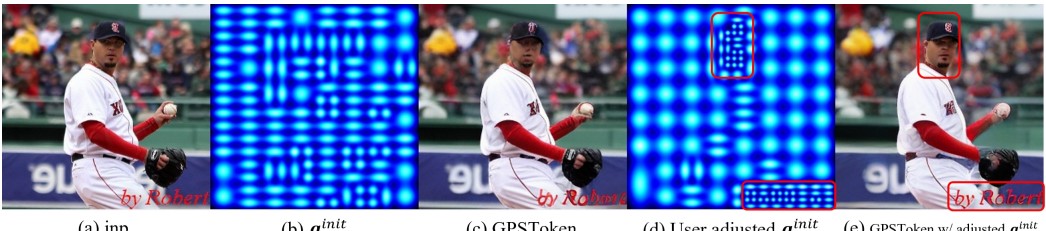

| (a) inp | (b) $\boldsymbol{g}^{init}$ | (c) GPSToken | (d) User adjusted $\boldsymbol{g}^{init}$ | (e) GPSToken w/ adjusted $\boldsymbol{g}^{init}$ |

Figure 6: **User-Controllable Adjustment of $\mathbf{g}^{init}$**. By manually setting $\mathbf{g}^{init}$, our GPSToken can focus more on semantically important regions (*e.g.* text and faces) and achieve finer reconstruction.

psnr: 19.26 ssim:0.600     psnr: 22.64 ssim:0.737     psnr: 26.03 ssim:0.834     psnr: 27.10 ssim:0.859

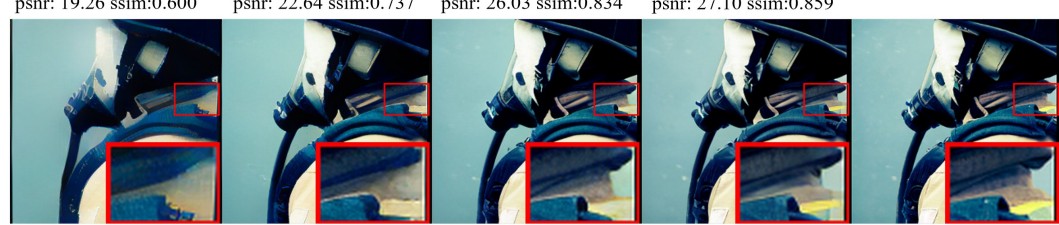

| 32 tokens | 64 tokens | 128 tokens (default) | 256 tokens | GT |

Figure 7: **Adjustable Token Count at Inference**. Token count can be adjusted at inference for better quality-efficiency trade-off, even beyond default training setting (128 tokens).

It can be clearly observed that with SA enabled, the maps exhibit significantly lower errors in complex regions (Figs. 5(f–g)), while the errors in simple regions (*e.g.*, background) remain largely unchanged. This demonstrates that our GPSToken improves the representation in complex areas without compromising the quality in simpler ones.

**User-Controllable Adjustment of $\mathbf{g}^{init}$.** Additionally, GPSToken supports manual adjustment of $\mathbf{g}^{init}$, allowing users to prioritize semantically important regions (*e.g.*, faces or text). An example is shown in Fig. 6, where placing denser Gaussians in target areas results in clearer reconstructions.

**Adjustable Token Count at Inference.** GPSToken supports adjustable token count at inference (see Fig. 7). Unlike [1, 24], which only supports decreasing the number of tokens at inference, GPSToken can also increase the number of tokens for improved quality.

Please refer to **Appendix** for more results of GPSToken on generalization performance, efficiency, ablation studies, and visualizations.

## 4.3 Image Generation

**Comparison Results.** We compare our approach against state-of-the-art tokenizers for image generation, including MaskGIT [3], TiTok [37], FlexTok [1], SoftVQ [6], ADM [8], One-D-Piece [24], DiT [26], SiT [23], REPA [38], D²iT [16], and MAETok [5], on $256 \times 256$ class-conditional generation tasks. Quantitative results with classifier-free guidance [14] are reported in Tab. 2. Our two-stage generator with 128 tokens outperforms all competing methods, achieving an FID of 1.50, highlighting the effectiveness of GPSToken in providing a superior latent space for generative models, even with fewer tokens. With an equal number of tokens (128), MAETok [5] under-performs our GPSToken-based generator, suggesting that our Gaussian-parameterized tokenization offers distinct advantages. In contrast, the one-stage generator slightly under-performs the baseline. This discrepancy arises from the optimization challenges inherent in the composition of Gaussian parameters $\mathbf{g}$ and textual features $\mathbf{f}$ within a single token $\mathbf{z}$. Our two-stage design (first generate $\mathbf{g}$ then synthesize $\mathbf{f}$), effectively addresses this issue, leading to significant enhancement over the baseline.

**Faster Training.** As illustrated in Fig. 8, our two-stage generator demonstrates significantly accelerated convergence compared to both the baseline and one-stage generator. Specifically, it achieves an FID-10K score of 25.48 within 100K iterations. In contrast, the SiT-XL/2 and one-stage generator reach scores of 25.41 and 26.20 after 300K and 500K iterations, respectively, indicating that our method is approximately $3\times$ and $5\times$ faster than them. This notable speed-up highlights the effectiveness of shape-texture decomposition in simplifying the optimization process. More results on training efficiency can be found in **Appendix**.

Table 2: Comparisons on $256 \times 256$ class-conditional image generation. The top 2 methods are highlighted in red and blue. "+" indicates the baseline.

| Method | Tokenizer | | Generator | |
|---|---|---|---|---|
| | Params (M) | Tokens | Params (M) | FID ↓ |
| **Auto-regressive Models** | | | | |
| MaskGIT [3] | 54.5 | $16\times16$ | 227 | 6.18 |
| FlexTok [1] | - | 256 | 1,330 | 2.02 |
| TiTok-S128 [37] | 83.7 | 128 | 287 | 1.97 |
| TiTok-B64 [37] | 204.8 | 64 | 177 | 2.77 |
| SoftVQ [6] | 173.6 | 64 | 675 | 1.78 |
| **Diffusion-based Models** | | | | |
| ADM [8] | - | - | 23.24 | 3.94 |
| One-D-Piece [24] | 83.9 | 256 | - | 2.35 |
| DiT-XL/2 [26] | 83.6 | $32\times32$ | 675 | 2.27 |
| SiT-XL/2+ [23] | 83.6 | $32\times32$ | 675 | 2.06 |
| REPA [38] | 83.6 | $32\times32$ | 675 | 1.79 |
| $D^2$iT [16] | - | >256 | 687 | 1.73 |
| MAETok [5] | 173.9 | 128 | 675 | 1.67 |
| **Ours (one-stage)** | 127.8 | 128 | 675 | 2.13 |
| **Ours (two-stage)** | 127.8 | 128 | 33+675 | 1.50 |

Figure 8: FID-10K training curves.

predicted $\boldsymbol{g}^{init}$ · calibrated $\boldsymbol{g}_{cal}^{init}$ · final $\boldsymbol{g}$ · synthesized image

Figure 9: Illustration of two-stage generation pipeline.

**Qualitative Analysis of Generation Process.** Fig. 9 provides a visual breakdown of the generation pipeline. Initially, $\mathbf{g}^{init}$ captures a coarse structure but may include incomplete or misaligned regions in Gaussian maps. Our calibration algorithm addresses these issues and refines $\mathbf{g}^{init}$ into a semantically coherent and spatially accurate layout $\mathbf{g}_{cal}^{init}$. Leveraging this calibrated layout, the second stage generates Gaussian parameters $\mathbf{g}$ that can encode local texture orientation and scale. Consequently, the final image not only retains the global structure established by $\mathbf{g}^{init}$ but also achieves rich details, exemplified by the synthesized dog/rabbit images. More results can be found in **Appendix**.

## 5 Conclusion

In this paper, we introduced **GPSToken**, a spatially-adaptive image tokenization approach for effective image representation and generation. Unlike conventional grid-based 2D/1D tokenizers, GPSToken leveraged parametric 2D Gaussian distributions to model image content in a non-uniform and content-aware manner. Our method achieved strong performance using only 128 tokens per image, yielding rFID and FID scores of 0.65 and 1.50 on image reconstruction and generation tasks, respectively. By decoupling spatial layout from texture features, GPSToken enabled a two-stage generation pipeline that supports flexible control over both structural and appearance attributes.

**Limitations**. While our approach has shown promising results, its heuristic initialization of Gaussian parameters may not always ensure optimal configurations. Future work could explore learning-based initialization to address this limitation. Additionally, designing a specialized architecture for predicting Gaussian parameters in generative tasks could improve layout synthesis and potentially eliminate the need for post-processing calibration, thereby enhancing overall performance.

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

# Appendix

In this appendix, we provide the following materials:

**A** Spatially-adaptive token initialization and Gaussian calibration algorithms (referring to Sec. 3.2 and Sec. 3.3 of the main paper);

**B** Experimental settings for reconstruction and generation (referring to Sec. 4.1 of the main paper);

**C** Generalization on higher resolutions and other datasets (referring to Sec. 4.2 of the main paper);

**D** FLOPs, memory and latency in reconstruction (referring to Sec. 4.2 of the main paper);

**E** Ablation studies on architecture designs and parameters (referring to Sec. 4.2 of the main paper);

**F** Training and inference efficiency of GPSToken generators (referring to Sec. 4.3 of the main paper);

**G** More visual comparisons and results on reconstruction and generative tasks (referring to Sec. 4.2 and Sec. 4.3 of the main paper);

**H** Broader impacts of GPSToken and its generator.

## A Spatially-adaptive Token Initialization and Gaussian Calibration

The spatially-adaptive token initialization algorithm is described in Sec. 3.1 of the main paper. It outlines a procedure for segmenting the entire image based on regional complexity and for initializing the Gaussian parameters of the GPS-tokens accordingly. The algorithm is summarized in Algorithm 1.

---

**Algorithm 1:** Spatially-adaptive Token Initialization Algorithm

**Input:** image $I$; target token count $l$; metric hyper-parameter $\lambda$; minimal size of region $s_{min}$.
**Output:** regions list $L$; initialized Gaussian parameters $\{\mathbf{g}_0^{init}, \cdots, \mathbf{g}_{l-1}^{init}\}$.

1 Initialize region candidate list $L = \{I\}$;
2 **while** $|L| < l$ **do**
3    Calculate complexities $\{m_0, \cdots, m_{|L|-1}\}$ for regions in $L$ using Eq. 4 in the main paper;
4    Let $\hat{L} = \{I_i \in L \mid \text{at least one side of } I_i \text{ is greater than } s_{\min}\}$;
5    Let $I_{\max} = \arg\max_{I_i \in \hat{L}} m_i$;
6    Obtain size $(w, h)$ of $I_{\max}$;
7    **if** $w \neq h$ **then**
8      Equally divide $I_{\max}$ into two sub-regions $I_1$ and $I_2$ along the longer side;
9      Update $L$ by replacing $I_{\max}$ with $\{I_1, I_2\}$;
10    **else**
11      **// step1: width-wise division -> compute complexities**
12      Divide $I_{\max}$ into $I_1, I_2$ along width;
13      Calculate complexities $m_1, m_2$ for $I_1, I_2$ using Eq. 4 in the main paper;
14      **// step2: hidth-wise division -> compute complexities**
15      Divide $I_{\max}$ into $I_3, I_4$ along height;
16      Calculate complexities $m_3, m_4$ for $I_3, I_4$ using Eq. 4 in the main paper;
17      **// step3: compare the complexities**
18      **if** $\min(m_1, m_2) \leq \min(m_3, m_4)$ **then**
19        Update $L$ by replacing $I_{\max}$ with $\{I_1, I_2\}$;
20      **else**
21        Update $L$ by replacing $I_{\max}$ with $\{I_3, I_4\}$;
22      **end**
23    **end**
24 **end**
25 Obtain size $(w_i, h_i)$ and center $(x_i, y_i)$ for each region $I_i \in L$;
26 Initialize $\mathbf{g}_i^{init} = \{\sigma_x^{(i)}, \sigma_y^{(i)}, \rho^{(i)}, \mu_x^{(i)}, \mu_y^{(i)}\} = \{\frac{w_i}{6}, \frac{h_i}{6}, 0, x_i, y_i\}$.

---

The Gaussian calibration algorithm is described in Sec. 3.2 of the main paper. It is used to refine the predicted Gaussians during the layout synthesis phase. The algorithm is summarized in Algorithm 2.

---

**Algorithm 2:** Gaussian Calibration Algorithm

---

**Input:** predicted Gaussian parameters $\{\mathbf{g}_0, \cdots, \mathbf{g}_{l-1}\}$; minimal size of region $s_{min}$; image size $W \times H$.
**Output:** calibrated Gaussian parameters $\{\mathbf{g}_0^{cal}, \cdots, \mathbf{g}_{l-1}^{cal}\}$.

1  **// step 1: calibrate the means** $(\mu_x, \mu_y)$
2  Let $\Delta x = \Delta y = s_{\min}$;
3  Define grid $\mathcal{G}$ as all points in $[0, W-1] \times [0, H-1]$ spaced by $(\Delta x, \Delta y)$;
4  Quantize each mean $(\mu_x^{(i)}, \mu_y^{(i)}) \in \mathbf{g}_i$ to the nearest central point in $\mathcal{G}$, obtaining $(\hat{\mu}_x^{(i)}, \hat{\mu}_y^{(i)})$;
5  **// step 2: re-initialize sigmas via region partitioning**
6  Initialize region list $L = \{(0, W, 0, H)\}$;
7  **while** $|L| < l$ **do**
8       Let $m_i$ be the count of quantized means in region $I_i \in L$;
9       Let $\hat{L} = \{I_i \in L \mid$ at least one side of $I_i$ is greater than $s_{\min}\}$;
10      Let $I_{\max} = \arg\max_{I_i \in \hat{L}} m_i$;
11      Let $x1, x2, y1, y2 \leftarrow I_{\max}$ and $(\hat{w}, \hat{h}) = (x2-x1, y2-y1)$;
12      **if** $\hat{w} > \hat{h}$ **then**
13          Replace $I_{\max}$ in $L$ with its two equal sub-regions split vertically along the x-axis;
14      **elif** $\hat{w} < \hat{h}$ **then**
15          Replace $I_{\max}$ in $L$ with its two equal sub-regions split horizontally along the y-axis;
16      **else**
17          **if** *no Gaussian falls exactly on the split line* $x = (x_1 + x_2)//2$ **then**
18              Replace $I_{\max}$ in $L$ with its two equal sub-regions split vertically along the x-axis;
19          **else**
20              Replace $I_{\max}$ in $L$ with its two equal sub-regions split horizontally along the y-axis;
21          **end**
22      **end**
23 **end**
24 Obtain size $(w_i, h_i)$ and center $(x_i, y_i)$ for each region $I_i \in L$;
25 Compute $\mathbf{g}_i^{cal} = \{\sigma_x^{cal-(i)}, \sigma_y^{cal-(i)}, \rho^{cal-(i)}, \mu_x^{cal-(i)}, \mu_y^{cal-(i)}\} = \{\frac{w_i}{6}, \frac{h_i}{6}, 0, x_i, y_i\}$.

---

## B  Experimental Settings

**Training and Inference Settings.** For the *image reconstruction task*, we train the encoder-decoder framework for 1M steps with a batch size of 96. The model is first trained using only the reconstruction loss $L_{\text{rec}}$ for the initial 600K steps. Subsequently, the perceptual loss $L_{\text{perc}}$ and adversarial loss $L_{\text{adv}}$ [11] are incorporated for the remaining 400K steps to enhance texture details. We use the Adam optimizer [19] with a fixed learning rate of $5 \times 10^{-5}$. Additionally, we apply an exponential moving average (EMA) with a decay rate of 0.9999 to stabilize the training process. We set $s = 5$ for Eq. 2 (in the main paper), $\lambda = 2.5$ for Eq. 4 (in the main paper) and $s_{min} = 4$ for Algorithm 1.

For the *image generation task*, we adopt the velocity matching loss from SiT [23] and train the layout and conditional texture generators sequentially. Specifically, the layout synthesis model is trained for 500K iterations, and the conditional layout-to-texture generation model for 4M iterations. To mitigate overfitting to the conditions, we add 0.5 Gaussian noise to the condition during training of the conditional texture generator. Both models are trained with a batch size of 256 and a learning rate of $1 \times 10^{-4}$ using the Adam optimizer. All experiments are conducted on eight A100 GPUs. During inference, we use a 5-step ODE sampler [23] to predict $\mathbf{g}^{\text{init}}$, followed by a 250-step SDE sampler [23], as used in SiT [23], for texture synthesis. We set the classifier-free guidance strength [14] to 1.5, following common practice. We set $s_{min} = 4$ for Algorithm 2.

**Network Architecture.** In the *image reconstruction task*, the encoder architecture comprises two residual blocks for extracting image features, followed by 30 transformation blocks designed to process initial Gaussian parameters and extract textual features for each region. During the rendering

Table 3: Comparisons of $512 \times 512$ and $1024 \times 1024$ reconstruction task on Imagenet val set.

| Method | Tokens | sample-level | | | distribution-level | |
|---|---|---|---|---|---|---|
| | | PSNR ↑ | SSIM ↑ | LPIPS ↓ | rec. FID ↓ | rec. sFID ↓ |
| $512 \times 512$ | | | | | | |
| SDXL-VAE [27] | 64×64 | 28.42 | 0.817 | 0.059 | 0.271 | 1.36 |
| VQVAE-f16 [11] | 32×32 | 21.83 | 0.604 | 0.172 | 2.29 | 7.95 |
| GPSToken-M128 | 512 | 26.74 | 0.764 | 0.073 | 0.367 | 1.93 |
| GPSToken-L256 | 1024 | 32.00 | 0.887 | 0.039 | 0.175 | 0.699 |
| $1024 \times 1024$ | | | | | | |
| SDXL-VAE [27] | 128×128 | 33.27 | 0.909 | 0.057 | 0.113 | 0.561 |
| VQVAE-f16 [11] | 64×64 | 25.41 | 0.744 | 0.169 | 1.40 | 4.98 |
| GPSToken-M128 | 2048 | 31.22 | 0.873 | 0.072 | 0.236 | 1.24 |
| GPSToken-L256 | 4096 | 37.71 | 0.955 | 0.031 | 0.055 | 0.276 |

stage, GPS-tokens are mapped into $64 \times 64$ 2D feature maps. The decoder adopts the same architecture as the last three stages of the SDXL-VAE [27] decoder but with double channels. GPSToken-M128 utilizes 128 tokens, each with 16 channels, whereas GPSToken-L256 employs 256 tokens, each with 32 channels, to match the capacity of VAVAE [36]. GPSToken-S64 uses only 64 tokens, each with 16 channels, but incorporates 60 transformation blocks within the encoder.

For the *image generation task*, we adopt the official SiT-S and SiT-XL architectures [23] as our backbone models. Specifically, SiT-S is utilized for layout synthesis, while SiT-XL is employed for conditional layout-to-texture generation. To support layout conditions, we use MLPs to transform these conditions before adding them into each attention block in SiT-XL.

## C  Generalization of GPSToken on Higher Resolutions and Other Datasets

**Higher Resolution.** We evaluate pre-trained GPSToken-M128 and GPSToken-L256 – originally trained on $256 \times 256$ images with 128 and 256 tokens, respectively – on $512 \times 512$ and $1024 \times 1024$ images from the ImageNet validation set. We compare with SDXL-VAE [27] and VQVAE-f16 [11], two public VAEs supporting reconstruction beyond training resolution. Following their practice, we scale the token count quadratically with resolution. For example, we use 512 and 2048 tokens for $512 \times 512$ and $1024 \times 1024$ inputs, respectively, when using GPSToken-M128.

As shown in Table 3, GPSToken shows strong generalization performance on resolution. Specifically, GPSToken-L256 achieves 32.00/0.887 (PSNR/SSIM) at $512 \times 512$ resolution and 37.71/0.955 at $1024 \times 1024$ resolution, outperforming SDXL-VAE in both settings. On the other hand, all methods show higher reconstruction performance at higher resolutions. GPSToken-M128 achieves a better "rec. FID" of 0.236 at $1024 \times 1024$ than that (0.367) at $512 \times 512$. This is because higher resolutions provide more pixels for the same content, increasing local redundancy and structural consistency. The denser pixel sampling makes fine details easier to recover, simplifying the reconstruction task despite the larger input size.

**More Datasets.** We further evaluate on additional datasets: COCO2017 [22] (natural images), FFHQ [17] (faces), STARE [15] (medical images), and WHU_RS19 [2] (remote sensing). We compare GPSToken with VAVAE [36] (256 tokens, 2D) and MAETok [5] (128 tokens, 1D), representing the state of the art in 2D and 1D tokenization, respectively. As shown in Table 4, GPSToken consistently outperforms both methods across all metrics and datasets under the same token counts.

Specifically, GPSToken-L256 achieves higher PSNR than VAVAE (256 tokens): 27.41 vs. 25.01 on COCO, 30.02 vs. 28.06 on FFHQ, 37.60 vs. 36.32 on STARE, and 26.33 vs. 23.57 on WHU_RS19. GPSToken-M128 also yields better LPIPS than MAETok (128 tokens): 0.083 vs. 0.101 on COCO, 0.050 vs. 0.064 on FFHQ, 0.036 vs. 0.051 on STARE, and 0.127 vs. 0.195 on WHU_RS19. These results show that GPSToken performs well not only on natural images but also on other domains – including medical and remote sensing – demonstrating strong generalization, robustness, and versatility for a wide range of vision tasks.

Table 4: Comparison of $256 \times 256$ image reconstruction on COCO, FFHQ, STARE, and WHU_RS19. For STARE and WHU_RS19, we only report PSNR, SSIM, and LPIPS, which are more appropriate for evaluating reconstruction quality on non-photorealistic images than metrics such as "rec. FID".

| Token Count | Method | PSNR ↑ | SSIM ↑ | LPIPS ↓ | rec. FID ↓ |
|---|---|---|---|---|---|
| **COCO** | | | | | |
| 128 | MAETok | 22.67 | 0.623 | 0.101 | 8.91 |
| | GPSToken-M128 | 23.47 | 0.657 | 0.083 | 4.72 |
| 256 | VAVAE | 25.01 | 0.736 | 0.052 | 6.01 |
| | GPSToken-L256 | 27.41 | 0.794 | 0.035 | 2.23 |
| **FFHQ** | | | | | |
| 128 | MAETok | 25.53 | 0.707 | 0.064 | 4.66 |
| | GPSToken-M128 | 26.35 | 0.745 | 0.050 | 3.72 |
| 256 | VAVAE | 28.06 | 0.808 | 0.027 | 1.95 |
| | GPSToken-L256 | 30.02 | 0.846 | 0.019 | 1.51 |
| **STARE** | | | | | |
| 128 | MAETok | 32.98 | 0.818 | 0.051 | - |
| | GPSToken-M128 | 34.75 | 0.868 | 0.036 | - |
| 256 | VAVAE | 36.32 | 0.896 | 0.019 | - |
| | GPSToken-L256 | 37.60 | 0.915 | 0.014 | - |
| **WHU_RS19** | | | | | |
| 128 | MAETok | 21.73 | 0.506 | 0.195 | - |
| | GPSToken-M128 | 23.20 | 0.560 | 0.127 | - |
| 256 | VAVAE | 23.57 | 0.619 | 0.142 | - |
| | GPSToken-L256 | 26.33 | 0.731 | 0.064 | - |

Table 5: Comparison of computational cost, memory usage, latency, and throughput for generating $256 \times 256$ images on an A100 GPU. Batch size is 8 for inference and 16 for training. Latency is averaged over 20 runs. Throughput is measured in samples per second. Training memory for FlexTok is unavailable due to the lack of released training code.

| Method | FLOPs (G) | | Memory (MB) | | Latency (ms) | Throughput (sample/s) |
|---|---|---|---|---|---|---|
| | Encoder | Decoder | Train | Inference | | |
| VQVAE-f16 | 556 | 1014 | 78222 | 2627 | 72 | 111.44 |
| TiTok-B64 | 220 | 973 | 45892 | 2359 | 96 | 83.44 |
| GPSToken-M128 | 383 | 2689 | 50793 | 2567 | 180 | 44.56 |
| GaussianToken | 1706 | 2285 | 60781 | 5352 | 181 | 44.32 |
| FlexTok | 283 | 7665 | – | 7275 | 2714 | 2.88 |

# D  FLOPs, Memory and Latency for Reconstruction Task

We provide profiling details on FLOPs, memory usage, latency, and throughput for the $256 \times 256$ image reconstruction task. As shown in Table 5, GPSToken incurs moderate computational cost. Although the FLOPs of GPSToken decoder are higher than those of VQVAE-f16 [33] and TiTok-B64 [37], its latency is competitive with GaussianToken [9] and significantly better than FlexTok [1], which employs a heavy autoregressive decoder. In terms of memory, GPSToken uses less GPU memory than FlexTok and GaussianToken during inference, and less training memory than VQVAE-f16, demonstrating favorable memory efficiency in both phases.

# E  Ablation Studies on Spatial Adaptivity Designs

**Ablation Studies on Components.** As described in Sec. 3.2 of the main paper, our GPSToken employs spatially-adaptive token initialization ("Init.") followed by spatially-adaptive token refinement

Table 6: Ablation studies of our spatial adaptivity designs on the $256 \times 256$ reconstruction task. ✓indicates that the component is used. "Init." and "Refine." denote the spatially-adaptive token initialization and spatially-adaptive token refinement, respectively.

| Method | Components | | sample-level | | | distribution-level | | | |
| --- | --- | --- | --- | --- | --- | --- | --- | --- | --- |
| | Init. | Refine. | PSNR ↑ | SSIM ↑ | LPIPS ↓ | rec. FID ↓ | rec. sFID ↓ | FID ↓ | sFID ↓ |
| **baseline** | | | 23.52 | 0.638 | 0.110 | 1.02 | 4.07 | 2.59 | 4.34 |
| **baseline+** | | ✓ | 24.00 | 0.654 | 0.100 | 0.81 | 3.59 | 2.37 | 4.31 |
| **GPSToken** | ✓ | ✓ | 24.06 | 0.657 | 0.080 | 0.65 | 3.28 | 2.18 | 3.96 |

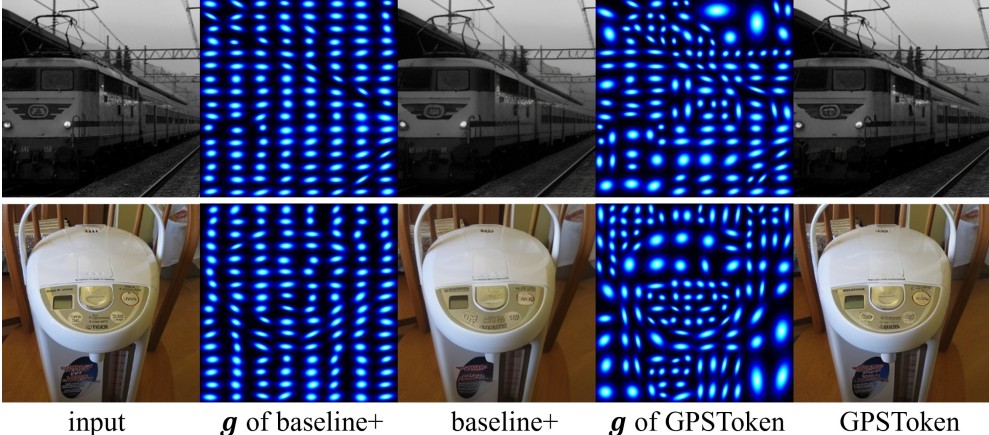

| input | $\boldsymbol{g}$ of baseline+ | baseline+ | $\boldsymbol{g}$ of GPSToken | GPSToken |

Figure 10: **Illustration of "baseline+" and GPSToken**. Left to right: the input image, visualization of Gaussians of "baseline+", the reconstruction of "baseline+", visualization of Gaussians of GPSToken, the reconstruction of GSPToken.

("Refine.") to progressively obtain coarse- and fine-grained spatial adaptations. We conduct ablation studies on GPSToken-M128 to validate the contribution of each component.

Table 6 presents the quantitative results. The baseline refers to the method that uses Gaussian-parameterized tokens without incorporating any spatially-adaptive components. The term "baseline+" denotes the method that additionally includes the "Refine." component. In contrast, GPSToken integrates both the "Init." and "Refine." components. As shown in the table, "baseline+" yields improvements over the baseline across both sample-level and distribution-level metrics, with a decrease of 0.01 in LPIPS and a decrease of 0.21 in "rec. FID". These enhancements indicate the general improvement achieved by adjusting Gaussians to match local textures. Compared to "baseline+", GPSToken significantly improves distribution-level metrics, achieving reductions of 0.19 in FID and 0.35 in sFID, while showing slight improvements in sample-level metrics (an increase of 0.06 in PSNR and 0.003 in SSIM). This demonstrates the effectiveness of "Init." component, which reallocates more Gaussians from simple regions to complex ones, thereby capturing finer semantic details in texture-rich regions without compromising reconstruction performance.

As shown in Fig. 10, without the "Init." component, the Gaussian maps from "baseline+" still roughly align in a 2D grid, even after refinement. This limits their ability to fit complex textures, only capturing edges in simple regions. In contrast, with the "Init." component, GPSToken aggregates more Gaussians in texture-rich areas, making it better suited to fit complex structures. This highlights the importance of "Init." component in achieving spatially adaptive representation of fine-grained visual contents.

**Ablation Studies on Params.** We further conduct experiments on the selection of parameters $\lambda$, $s$, and $s_{\min}$. The results are shown in Table 7.

- **Entropy Threshold** $\lambda$: As stated in Eq. 4 of the main paper, $\lambda$ balances region size and complexity. A larger $\lambda = 5$ encourages Gaussians to concentrate on complex regions, leading to only minor performance degradation. In contrast, setting $\lambda = 0$ allocates Gaussians solely based on region size, resulting in a uniform spatial distribution. This causes a significant drop in performance:

Table 7: Hyper-parameter selection on $\lambda$, $s$, and $s_{\min}$.

| Hyper-parameter | | PSNR | SSIM | LPIPS | rec. FID | FID |
|---|---|---|---|---|---|---|
| $\lambda$ | 0 | 23.52 | 0.638 | 0.110 | 1.02 | 2.59 |
| | 5 | 24.06 | 0.652 | 0.083 | 0.68 | 2.23 |
| $s$ | 1 | 17.55 | 0.439 | 0.344 | 160 | 165 |
| | 3 | 24.07 | 0.657 | 0.080 | 0.66 | 2.18 |
| | 7 | 24.06 | 0.658 | 0.080 | 0.66 | 2.16 |
| $s_{\min}$ | 8 | 24.05 | 0.656 | 0.080 | 0.66 | 2.18 |
| | 16 | 24.03 | 0.657 | 0.081 | 0.66 | 2.19 |
| **ours** ($\lambda = 2.5, s = 5, s_{\min} = 4$) | | 24.06 | 0.657 | 0.080 | 0.65 | 2.18 |

Table 8: Training and Inference Efficiency Comparison between SiT-XL/2 Baseline and GPSToken generator on ImageNet 256×256 with A100 GPU.

| Method | Metric | 500K | 1000K | T-Mem | T-Thpt | I-Mem | I-Thpt |
|---|---|---|---|---|---|---|---|
| Baseline | FID | 19.07 | 14.50 | 63684 | 0.63 | 9126 | 0.067 |
| | Time (h) | 219 | 439 | | | | |
| Ours | FID | 9.57 | 7.61 | 41498 | 1.09 | 9636 | 0.129 |
| | Time (h) | 128 | 256 | | | | |

**Notes:** T-Mem: Training Memory (MB), T-Thpt: Training Throughput (iters/s), I-Mem: Inference Memory (MB), I-Thpt: Inference Throughput (samples/s)

LPIPS increases from 0.08 to 0.11, and rec.FID rises from 0.65 to 1.02. We set $\lambda$ to 2.5 based on experimental experience.

- **Support Factor** $s$: As stated in Eq. 2 of the main paper, $s$ controls the effective rendering support of each Gaussian. Performance degrades significantly when $s = 1$, but stabilizes for $s \geq 3$. This aligns with the $3\sigma$ rule, *i.e.*, 99.7% of the mass of a 2D Gaussian lies within three standard deviations from the mean. To ensure full coverage, we set $s = 5$ in all experiments.

- **Minimal Region Size** $s_{min}$: We set $s_{min}$ in the calibration algorithm to match its value in the initialization algorithm, where $s_{min}$ determines the minimum width or height of each region. We observe that increasing $s_{min}$ from 4 to 16 results in negligible performance degradation. This is expected because, with 128 tokens representing a 256×256 image, the average spatial extent per token is approximately 22×22 pixels. Consequently, most of the segmented regions naturally have a width or height greater than or equal to 16, making the choice of $s_{min}$ within this range largely inconsequential for the final tokenization.

## F   Training/Inference Efficiency of GPSToken Generators

We provide comprehensive computational benchmarks comparing the GPSToken generator with the SiT-XL/2 baseline in both training and inference, as shown in Table 8. At 1M iterations, GPSToken achieves a significantly lower FID score (7.61 vs. 14.50), with 42% less training time (256h vs. 439h), 73% higher training throughput (1.09 vs. 0.63 iters/s), and 35% lower VRAM consumption (41,498 MB vs. 63,684 MB). During inference, although VRAM usage increases slightly (9,636 MB vs. 9,126 MB), our method nearly doubles the throughput (0.129 vs. 0.067 samples/s), reducing latency by approximately half.

These efficiency gains stem from two key design choices: (i) GPSToken reduces the number of effective tokens, lowering computational and memory overhead; (ii) the two-stage generation framework simplifies the learning objective and stabilizes optimization, enabling faster convergence to higher-quality solutions. Overall, GPSToken not only improves generation quality but also substantially reduces training cost and inference latency.

# G  More Visual Results

## G.1  Results for Reconstruction Task

**Visual Comparisons.** We provide visual comparisons among GPSToken and its competitors in Figs. 11 and 12. It can be observed that our GPSToken achieves significantly more accurate and clearer textures in complex regions, without compromising the performance in simpler areas.

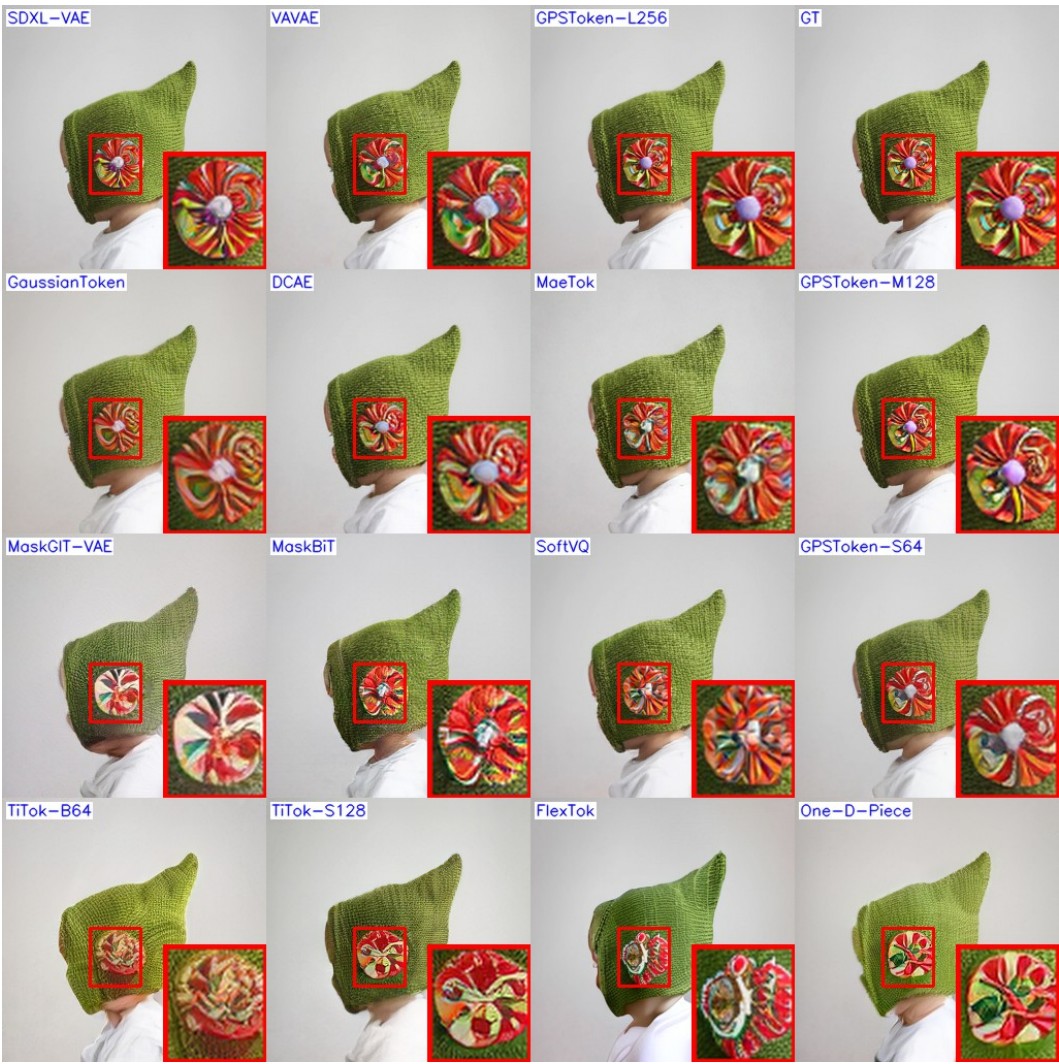

Figure 11: **Visual comparisons on** $256 \times 256$ **reconstruction task.**

**More Visual Results.** Further visual results of our spatially adaptive designs are presented in Fig. 13. Fig. 14 illustrates the adjustment of the initial Gaussian parameters $\mathbf{g}^{\text{init}}$ to better focus on the regions of interest. Fig. 15 shows the flexibility to adjust token counts during inference, demonstrating the adaptability of our approach under varying length.

## G.2  Results for Generation Task

Fig. 16 shows a few images generated by our two-stage generator. One can see that our generator is capable of synthesizing natural images depicting a wide variety of scenes. For instance, it successfully generates fine details in objects such as beetles, eagles, trucks, bags, mailboxes, golf balls, dinosaur fossils, and so on. Furthermore, the generated images exhibit high visual quality - characterized by

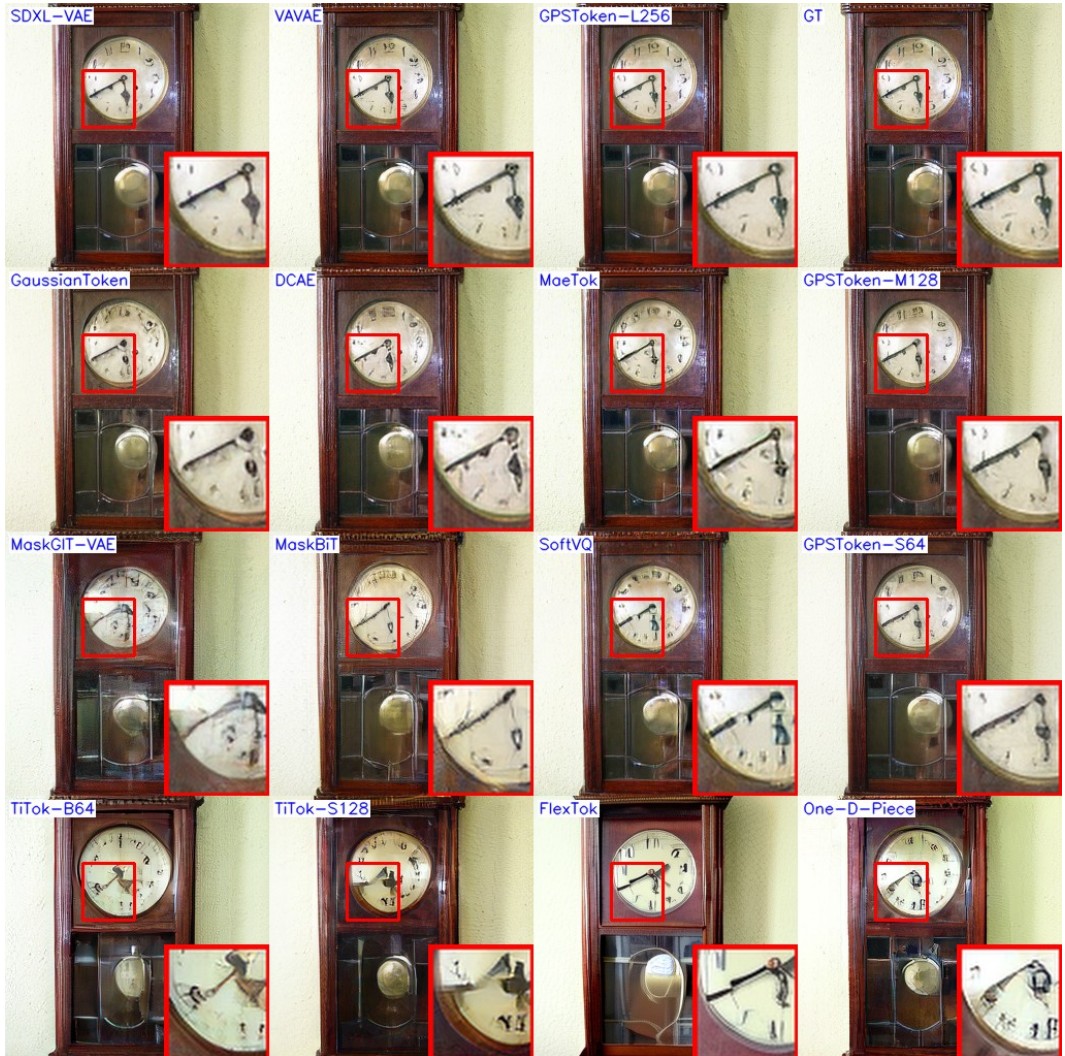

Figure 12: **Visual comparisons on** $256 \times 256$ **reconstruction task.**

sharp details and realistic textures - demonstrating the generator's strong ability to synthesize diverse and photo-realistic images.

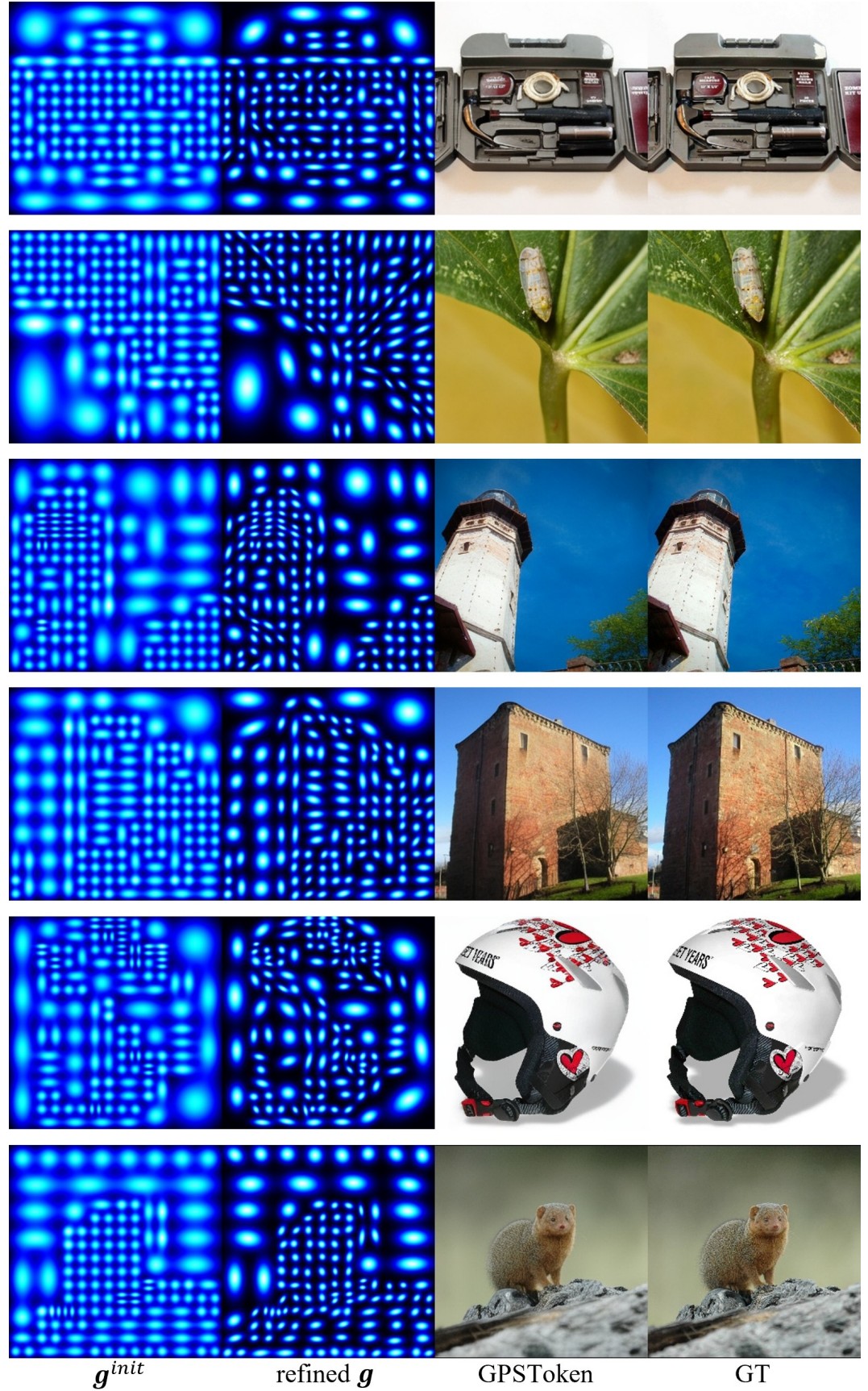

$g^{init}$    refined $g$    GPSToken    GT

Figure 13: **More visual results of spatial adaptivity.**

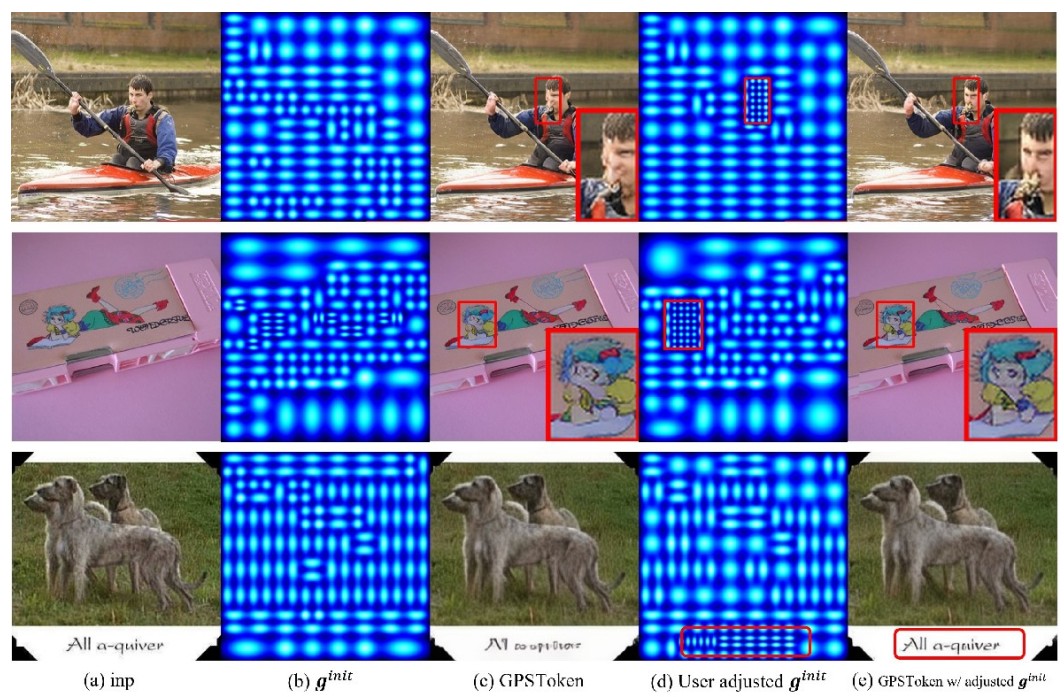

| (a) inp | (b) $\boldsymbol{g}^{init}$ | (c) GPSToken | (d) User adjusted $\boldsymbol{g}^{init}$ | (e) GPSToken w/ adjusted $\boldsymbol{g}^{init}$ |

Figure 14: **More visual results on User-Controllable Adjustment of $\mathbf{g}^{init}$.**

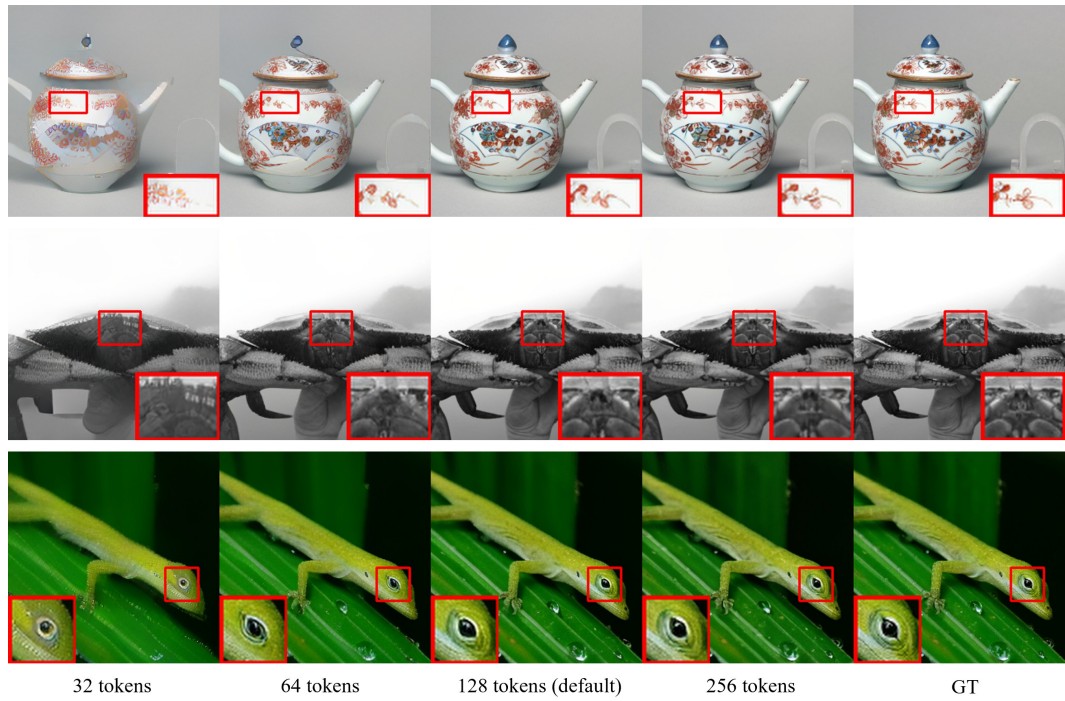

| 32 tokens | 64 tokens | 128 tokens (default) | 256 tokens | GT |

Figure 15: **More visual results on Adjustment of Token Count at Inference.**

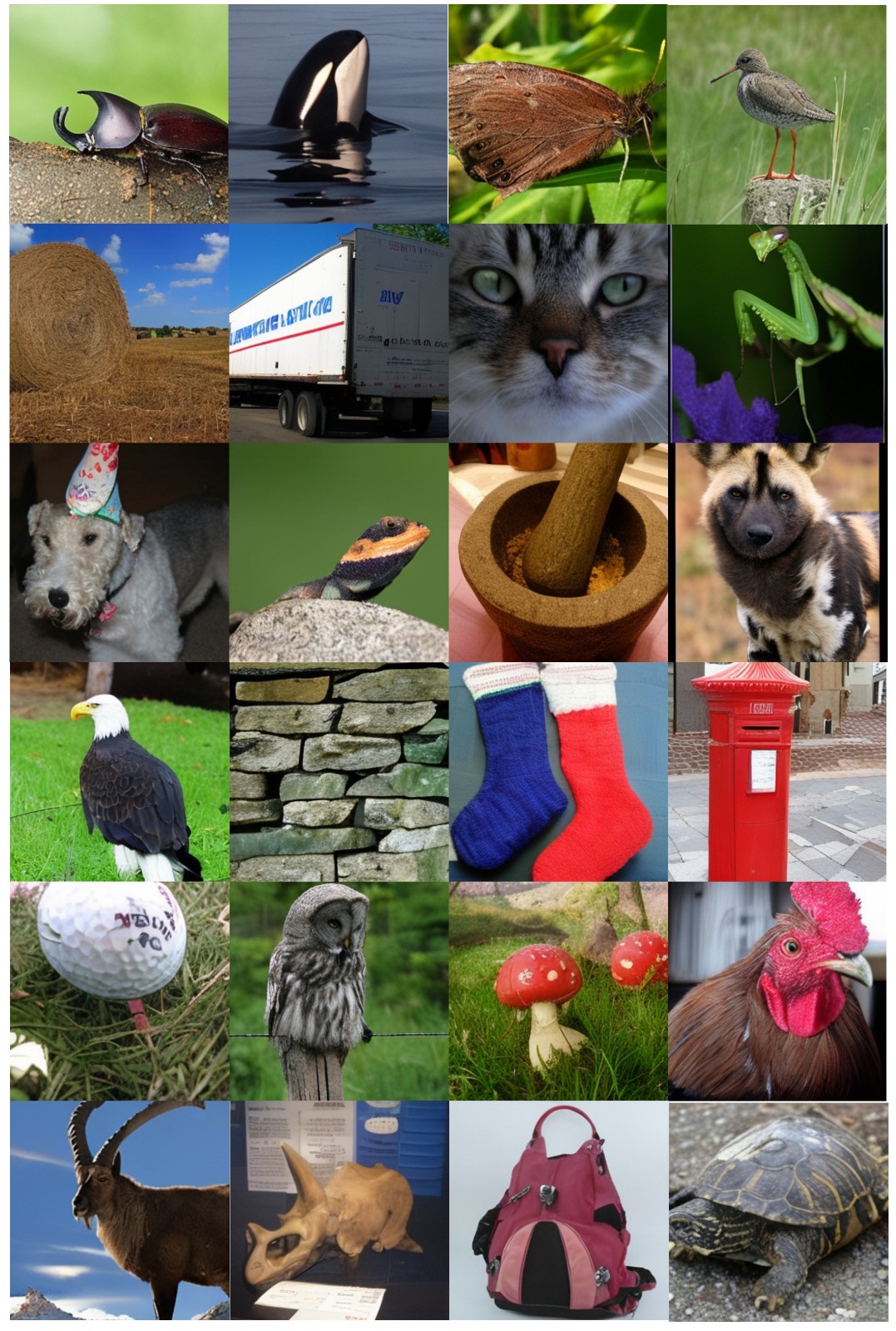

Figure 16: **Visual result of** $256 \times 256$ **generation.**

## H Broader Impacts

This work introduces GPSToken, a spatially-adaptive tokenization framework designed to enable efficient and content-aware image representation. By offering flexible feature modeling, GPSToken enhances representational capacity, benefiting both computer vision researchers and downstream applications in domains such as medical imaging and creative design. Furthermore, its two-stage layout-texture synthesis approach reduces computational barriers for generative tasks, making it accessible to individual users and small companies.

Despite its potential, the deployment of GPSToken also presents several risks. The ability to generate high-quality synthetic media may be misused, potentially harming vulnerable populations through the spread of misinformation or deepfake technologies. Additionally, if trained on biased datasets, the model may encode disparities in texture and shape representation, which could compromise fairness - particularly in sensitive applications such as facial recognition. In safety-critical domains like autonomous driving or medical diagnosis, failures in accurate tokenization could lead to misinterpretation of complex visual scenes, with potentially dangerous consequences.

