# OpenReview forum: "GPSToken: Gaussian Parameterized Spatially-adaptive Tokenization for Image Representation and Generation"
_NeurIPS.cc/2025/Conference — NeurIPS 2025 poster_

### Official Review · Reviewer_pm2Q · 2025-06-28

**Clarity:** 3
**Significance:** 2
**Originality:** 3
**Rating:** 4
**Confidence:** 2

**Summary:**

This paper introduces GPSToken, a Gaussian Parameterized Spatially-adaptive Tokenization framework for image representation and generation.
Unlike conventional grid-based tokenizers, GPSToken uses parametric 2D Gaussians to dynamically model regions of varying shapes, positions, and textures, enabling non-uniform and content-aware tokenization.
The proposed method partitions images into texture-homogeneous regions, initializes Gaussian parameters based on entropy-driven complexity, and refines them via a transformer.
 A splatting-based renderer converts tokens back into feature maps for decoding. GPSToken decouples spatial layout (Gaussian parameters) from texture features, facilitating a two-stage generation pipeline that first synthesizes structural layouts and then generates textures.
 Experiments on ImageNet demonstrate state-of-the-art performance.

**Questions:**

1. In tab.2, I notice that the proposed method and MAETok have similar performance. However, the proposed method has a larger generator, and the MAETok has a larger tokenizer. Which model part do you think will influence the performance most? Are there any experimental results that can reflect this?

**Ethical Concerns:**

["NO or VERY MINOR ethics concerns only"]

**Final Justification:**

Overall, I think leveraging parametric 2D Gaussians to model different image regions is a very interesting idea, and the authors also provide a reasonable pipeline to realize the proposed idea.
While I initially had some concerns about the experiments, the authors addressed them satisfactorily in their rebuttal.
Thus, I maintain my positive score.

**Limitations:**

Yes

**Quality:**

3

**Strengths And Weaknesses:**

# Strengths

1. The main idea of this paper is quite interesting, and the proposed methods are reasonable.

2. The writing of this paper is easy to follow.


# Weaknesses

1. The experiments seem unfair in some cases. For example, in Tab.1, the parameter size of the VAVAE is only 69.8M, which is significantly smaller than the proposed method.

2. I think it will be better to completely discuss the training overhead of the proposed method and compare it with existing SOTA methods instead of only discussing SiT-XL/2.

I am not an expert in this field, and I may not understand all the strengths and weaknesses of this paper.

---

> ### Author Rebuttal · Authors · 2025-07-30
>
> We sincerely thank this reviewer for the constructive comments and suggestion. We hope our following point-to-point responses can address this reviewer's concerns.
>
> ---
>
> **Q1: Parameter size**.
>
> We thank the reviewer for the insightful comment. Although VAVAE has fewer parameters (69.8M) than our method (128M), our model size is moderate among the 14 methods compared in Tab. 1 of the main paper. Moreover, direct comparison based solely on parameter count is not reasonable due to the significant architectural and training differences: SDXL-VAE uses residual blocks, TiTok employs transformers, VAVAE leverages a large pretrained vision model, FlexTok incorporates a generative decoder, and our GPSToken introduces Gaussians as a novel image representation. Notably, FlexTok has about eight times the number of parameters as our GPSToken, yet it performs significantly worse — demonstrating that parameter count alone does not determine performance.
>
> More importantly, the number of tokens has a greater impact on the downstream generator than the tokenizer's parameter count, as it directly affects the optimization difficulty and the computational cost—note that the generator typically contains far more parameters (e.g., 708M in our case). Consequently, most prior works focus on improving reconstruction quality under the same token count, placing less emphasis on tokenizer parameter efficiency. In Tab. 1 of the main paper, under this standard and fair setting (same token count), our GPSToken achieves the best reconstruction performance across all metrics.
>
> We will make these points clearer in the revision.
>
> ---
>
> **Q2: Training overhead**.
>
> We sincerely thank the reviewer for the insightful comment. We agree that a comprehensive discussion on the training overhead in comparison with existing SOTA methods is valuable.
>
> However, due to the substantial computational cost—our SiT-XL/2 baseline takes approximately 9 days for 400k iterations on our GPU cluster—conducting extensive retraining across multiple SOTA frameworks within the rebuttal period is unfortunately infeasible.
>
> Instead, we can provide an indirect comparison based on the reported results in literature. As stated in their papers, REPA achieves comparable performance to SiT-XL/2 (400k iters) in just 100k iterations, and MAETok reaches similar performance in about 50k iterations. In contrast, our GPSToken matches the 400k-iteration performance of SiT-XL/2 in approximately 150k iterations. Thus, in terms of convergence speed measured by iteration count, REPA and MAETok converge faster than our two-stage framework.
>
> However, per-iteration efficiency and wall-clock training time are equally important. REPA relies on a large pretrained vision model (DINOv2-L) on the base of SiT-XL/2, leading to high computational and memory overhead per step. In contrast, our method significantly improves per-step efficiency than SiT-XL/2 (see our response to Q2 of reviewer xDKG). Therefore, despite requiring more iterations, our approach achieves competitive or even superior wall-clock training speed in practice.
>
> It should be noted that our method is orthogonal to existing acceleration techniques, such as auxiliary losses (e.g., in MAETok, REPA). We believe these techniques can be integrated into our framework for even greater efficiency. We also would like to clarify that the primary contribution of GPSToken is a novel Gaussian-based visual tokenization method and a two-stage generation framework, enabling effective image representation and structured generation. Reduced training overhead is a beneficial byproduct, not our main focus.
>
> We will revise the manuscript to include a more comprehensive discussion on these aspects.
>
> ---
>
> **Q3: Tokenizer vs. generator**.
>
> Thank you for the insightful observation. The difference in model scale—our GPSToken generator is slightly larger while MAETok employs a relatively larger tokenizer—stems primarily from the disparate experimental setups and architectural choices across methods, rather than intentional design for performance tuning. Notably, our approach employs a conventional residual block-based decoder, whereas MAETok utilizes a transformer-based decoder. These fundamental architectural distinctions naturally lead to variations in parameter distribution and model size.
>
> Regarding which component—tokenizer or generator—has a greater influence on overall performance, this remains an open question in the field. Prior work exhibits diverse emphases: methods such as VQVAE and TiTok focus heavily on improving tokenizer fidelity under the premise that better reconstruction leads to stronger generative performance. In contrast, approaches such as REPA employ auxiliary losses to guide generator training, effectively treating the tokenizer as a secondary component. More recently, emerging frameworks such as FlexTok and REPA-E advocate for joint optimization or unified modeling of tokenization and generation, blurring the boundary between the two modules. This evolving landscape makes it challenging to isolate the relative importance of each part under the current paradigms.
>
> In our view, the tokenizer plays a more critical role in the overall system. It shapes the structure of the latent space, thereby determining the intrinsic complexity of the generative task and establishing the theoretical performance ceiling for the generator. A poorly structured latent space can impose optimization barriers that even a powerful generator may struggle to overcome.
>
> By far, there lack comprehensive empirical studies to systematically ablate or compare the contributions of tokenizers and generators under controlled settings. This gap is largely due to the significant computational resources required and the difficulty in establishing fair comparisons across diverse architectures, training objectives, and theoretical foundations. We believe that developing standardized benchmarks for modular evaluation—such as plug-and-play tests between tokenizers and generators—would be a valuable direction for future work.

---

> ### Author Response · Authors · 2025-08-04
>
> Dear Reviewer pm2Q,
>
> Many thanks for your time in reviewing our paper and your constructive comments. We have submitted the point-to-point responses. We appreciate if you could let us know whether your concerns have been addressed, and we are happy to answer any further questions.
>
> Best regards,
>
> Authors of paper \#1542

---

### Official Review · Reviewer_xDKG · 2025-07-03

**Clarity:** 2
**Significance:** 3
**Originality:** 3
**Rating:** 5
**Confidence:** 4

**Summary:**

GPSToken proposes a Gaussian parameterized spatially-adaptive tokenization framework to address the inflexibility of conventional 2D/1D grid tokenization in representing regions with varying shapes, textures, and locations. The framework achieves non-uniform image tokenization by: 1) partitioning images into texture-homogeneous regions using an entropy-driven algorithm; 2) parameterizing each region as a 2D Gaussian (mean for position, covariance for shape) with texture features; 3) optimizing Gaussian parameters via a specialized Transformer for continuous adaptation and content-aware feature extraction. During decoding, Gaussian-parameterized tokens are reconstructed into 2D feature maps using a differentiable splatting-based renderer, enabling end-to-end training with standard decoders. By decoupling spatial layout (Gaussian parameters) from texture features, GPSToken enables a two-stage generation pipeline: structural layout synthesis via lightweight networks, followed by structure-conditioned texture generation.

**Questions:**

1. In the "GPSToken-driven Two-stage Image Generation" framework, which specific loss function is employed to optimize the model, and how do these losses contribute to the overall generation quality and semantic consistency of the generated images?
2. What are the detailed training recipe adopted in "GPSToken-driven Two-stage Image Generation"? For instance, are there any specific layers or components, such as the decoder, whose weights are frozen during certain training phases? And how do these strategies impact the model's convergence speed and performance in generating high-fidelity images?

**Ethical Concerns:**

["NO or VERY MINOR ethics concerns only"]

**Final Justification:**

Thanks for the response from the authors. The rebuttal has addressed my concerns. I hope the authors include the additional experiments in their manuscript to better support their claims.

**Limitations:**

yes

**Paper Formatting Concerns:**

No major concerns

**Quality:**

3

**Strengths And Weaknesses:**

Strength:
1. Unlike prior grid-based or 1D tokenization, GPSToken introduces a continuous, parametric representation of image regions using 2D Gaussians, which dynamically adapts to local structures. This is a departure from discrete or fixed-grid methods.
2. The two-stage generation pipeline (structure-first, texture-later) aligns with human perception and reduces computational complexity, making it feasible for real-world applications. The ability to adjust token count at inference enhances flexibility for quality-efficiency trade-offs.
3. The paper outlines the entropy-driven partitioning algorithm, Gaussian parameterization, and transformer refinement process in sufficient detail, including mathematical formulations (e.g., Eq. 2 for modified Gaussian, Eq. 3 for splatting). Experimental settings (dataset, tr-aining configurations, metrics) are well-documented.
4. The paper provides extensive quantitative results on ImageNet, comparing against a wide range of state-of-the-art tokenizers (e.g., SDXL-VAE, TiTok, MAETok) across reconstruction and generation tasks. The performance gains (e.g., FID 1.64 with 128 tokens) are significant and supported by multiple metrics (PSNR, SSIM, LPIPS).

Weakness:
1.	Experiments are primarily conducted on ImageNet 256*256. More experiments on ImageNet 512*512 or t2i could be conducted.
2.	The paper highlights faster training convergence in line 63 but lacks concrete metrics (e.g., GPU hours, memory footprint) to quantify computational efficiency against baselines (e.g., SiT-XL/2). How does GPSToken optimize practical deployment costs? Please supplement with detailed computational benchmarks (e.g., training time per epoch, inference time, VRAM usage) and comparative analysis.

---

> ### Author Rebuttal · Authors · 2025-07-30
>
> We sincerely thank this reviewer for the constructive comments and suggestion. We hope our following point-to-point responses can address this reviewer's concerns.
>
> ---
>
> **Q1: Higher resolution and T2I tasks**.
>
> Regarding higher-resolution image reconstruction tasks, we refer the reviewer to our response to Q4 of Reviewer EBwp, where additional experiments on $512\times 512$ and $1024\times 1024$ images are presented. The successful tokenization and faithful reconstruction at these scales indicate that our method can reliably encode fine-grained visual details -- a critical prerequisite for high-resolution generation.
>
> Regarding high-resolution or text-to-image (T2I) generation tasks, we acknowledge that additional experiments would further validate the applicability of our tokenizer in generative frameworks. However, such experiments generally require substantial computational resources, typically involving several weeks to months of training on large GPU clusters. Due to these practical constraints, we are currently unable to complete them within the rebuttal timeline. We plan to conduct these studies as part of our future work.
>
> We believe that our experiments are well-aligned with recent advances in the context of image tokenizers. Many recent works (e.g., MaskBit (ICLR 2025), VAVAE (CVPR 2025)) primarily focus on reconstruction and generation tasks on ImageNet $256\times 256$ images. Our experimental design follows this established convention, prioritizing the analysis of the tokenization process itself.
>
> ---
>
> **Q2: Computational benchmarks**
>
> We thank the reviewer for the insightful comment. In response, we provide comprehensive computational benchmarks by comparing GPSToken with the SiT-XL/2 baseline in both training and inference.
>
> As shown in the table below, at 1M iterations, GPSToken achieves a significantly lower FID score (7.61 vs. 14.50), with 42% less training time (256h vs. 439h), 73% higher training throughput (1.09 vs. 0.63 iters/s), and 35% lower VRAM consumption (41,498 MB vs. 63,684 MB). During inference, although VRAM usage increases slightly (9,636 MB vs. 9,126 MB), our method nearly doubles throughput (0.129 vs. 0.067 samples/s), reducing latency by approximately half.
>
> | Method  | Metric     | 500K    | 1000K   | T-Mem (MB) | T-Thpt (iters/s) | I-Mem (MB) | I-Thpt (samples/s) |
> |---------|------------|---------|--------|------------|------------------|------------|--------------------|
> | Baseline | FID        | 19.07   | 14.50  | 63,684     | 0.63             | 9,126      | 0.067              |
> |         | Time (h)   | 219     | 439    |            |                  |            |                    |
> | Ours     | FID        | 9.57    | 7.61   | 41,498     | 1.09             | 9,636      | 0.129              |
> |         | Time (h)   | 128     | 256    |            |                  |            |                    |
>
> *Notes: T-Mem: Training Memory, T-Thpt: Training Throughput, I-Mem: Inference Memory, I-Thpt: Inference Throughput*
>
> These efficiency gains are attributed to two key design choices:
>
> (i) GPSToken reduces the number of effective tokens, lowering computational and memory overhead;
> (ii) the two-stage generation framework simplifies the learning objective and stabilizes optimization, facilitating faster convergence to higher-quality solutions.
>
> Overall, GPSToken not only improves generation quality but also significantly reduces training cost and inference latency.
>
> ---
>
> **Q3: Loss functions for generators**.
>
> Thank you for the question. We apologize for the lack of details in loss function for generators.
>
> Both the layout and texture generators use the standard **velocity matching loss** from SiT: the model predicts the diffusion velocity $\mathbf{v}_\theta(\mathbf{x}_t, t)$, and we apply an $L_2$ loss to match the target $\mathbf{v}^*$:
>
> $$
> \mathcal{L} = \mathbb{E} \left[ \|\mathbf{v}_\theta(\mathbf{x}_t, t) - \mathbf{v}^*\|^2 \right].
> $$
>
> This loss promotes high generation quality by modeling data dynamics accurately. In the layout stage, it encourages semantically coherent Gaussian arrangements. In the texture stage, it ensures fidelity and alignment with layout conditions, enhancing overall semantic consistency.
>
> ---
>
> **Q4: Training recipe for generators.**
>
> Both the layout and texture generators are trained **independently and end-to-end**, with **no component frozen** — all parameters are updated during training.
>
> Specifically, the **layout generator** uses a SiT-B/4 architecture and is trained to model $g_{init}$ (which encodes only the position and shape of initialized Gaussians) using the standard velocity matching loss. Since $g_{init}$ represents a simple geometric prior, its distribution is relatively easy to learn. The **conditional texture generator** employs a SiT-XL/2 architecture and learns to generate the refined layout $g$ and image features $f$, using $\{g, f\}$ as targets and the same velocity matching loss. Although modeling their joint distribution is challenging, conditioning on the generated $g_{init}$ — from which $g$ is refined — provides strong structural guidance, significantly reducing optimization difficulty.
>
> At inference, a **calibration step** is employed to refine $g_{init}$ to correct minor errors (L190 in the main paper), mitigating potential misalignment from independent training.
>
> As shown in Tab. 2 and Figs. 7/8 in the main paper, this strategy enables faster convergence and high-fidelity generation, validating the effectiveness of our decoupled yet coherent two-stage design.

---

> > ### Comment · Reviewer_xDKG · 2025-08-08
> >
> > I appreciate the authors' detailed response. All my concerns have been resolved, and I will raise my score.

---

> > > ### Author Response · Authors · 2025-08-09
> > >
> > > Many thanks for the support on our work! We will adopt your suggestions into the revision of the manuscript.
> > >
> > > Authors of paper #1542

---

> ### Author Response · Authors · 2025-08-04
>
> Dear Reviewer xDKG,
>
> Many thanks for your time in reviewing our paper and your constructive comments. We have submitted the point-to-point responses. We appreciate if you could let us know whether your concerns have been addressed, and we are happy to answer any further questions.
>
> Best regards,
>
> Authors of paper \#1542

---

### Official Review · Reviewer_pWj1 · 2025-07-03

**Clarity:** 3
**Significance:** 3
**Originality:** 3
**Rating:** 4
**Confidence:** 3

**Summary:**

This paper addresses the limitations of conventional grid-based image tokenizers, which lack spatial adaptivity and content-awareness. The authors propose GPSToken, a Gaussian-parameterized tokenization framework that dynamically models image regions using 2D Gaussians to encode their shape, position, and texture. By disentangling layout from appearance, GPSToken enables a two-stage image generation pipeline—first synthesizing structural layouts, then generating textures. Experiments on ImageNet show that GPSToken achieves state-of-the-art performance on both image reconstruction and generation tasks, with significantly fewer tokens and faster convergence.

**Questions:**

1.	What is the basis for setting ${\sigma }_{x}^{\left( i\right) }$ and ${\sigma }_{y}^{\left( i\right) }$ to $\frac{1}{6}$ for ${w}_{i}$ and ${h}_{i}$ to ensure full coverage during rendering ? Is this choice empirically optimal? How sensitive are the results to this initialization?

2.	How does the method perform on real-world datasets beyond ImageNet? Any plans to test on more diverse datasets (e.g., COCO, FFHQ)?？

**Ethical Concerns:**

["NO or VERY MINOR ethics concerns only"]

**Limitations:**

The authors have briefly acknowledged the heuristic limitation in Gaussian initialization and the need for more specialized generation architecture. However, it would be helpful to explicitly discuss:

1.	Potential failure cases (e.g., cluttered scenes, occlusions).
2.	How token count influences spatial bias or missed details.
3.	Scalability to higher resolutions.

**Quality:**

3

**Strengths And Weaknesses:**

The research problem is meaningful, and the authors present their ideas clearly, with careful and thorough explanation of the modeling process. The writing is solid and communicates the technical components effectively.

However, a key weakness is that while the core idea—using 2D Gaussians to dynamically model variable region shapes and positions—is appealing in reducing redundancy in simple regions and enabling finer granularity in complex ones. The motivation for introducing 2D Gaussian functions is not clearly established, as the paper lacks a deeper analysis connecting their specific properties to the concrete demands of the tokenization or generation tasks. I would encourage the authors to further articulate the theoretical motivation and necessity for using this specific formulation.

In addition, the experimental section seems to lack necessary ablation studies to justify design choices, and the setup feels somewhat underexplored. Stronger empirical evidence would improve the credibility and completeness of the work.

---

> ### Author Rebuttal · Authors · 2025-07-30
>
> We sincerely thank this reviewer for the constructive comments and suggestion. We hope our following point-to-point responses can address this reviewer's concerns.
>
> ---
>
> **Q1: Motivation for 2D Gaussians in tokenization.**
>
> Our choice of 2D Gaussian functions for image region modeling is grounded in their mathematical properties, which align well with the requirements of adaptive image tokenization.
>
> Standard grid-based tokenization methods suffer from spatial redundancy—simple regions are over-partitioned, while complex ones lack sufficient resolution.  An ideal tokenization scheme should have the following properties:
>
> (1) flexibly represent semantic regions of arbitrary location and scale;
> (2) model fuzzy boundaries in a soft, probabilistic manner;
> (3) be fully differentiable to enable end-to-end optimization with gradient-based learning; and
> (4) maintain low parametric complexity to ease downstream generation tasks.
>
> After evaluating alternatives such as bounding boxes and segmentation masks, we find that 2D Gaussians offer a principled and effective solution:
>
> - **Spatial adaptivity:** Each 2D Gaussian is parameterized by a mean $\mu = (\mu_x, \mu_y)$ and a covariance matrix (determined by $\sigma_x$, $\sigma_y$, and $\rho$), enabling anisotropic shapes that adapt to region geometry. Multiple Gaussians can be combined to model complex structures.
>
> - **Soft boundary modeling:** The Gaussian density function provides a smooth, continuous weight distribution, naturally capturing the uncertainty and gradual transitions at object boundaries in natural images.
>
> - **End-to-end differentiability:** All parameters are differentiable, and feature aggregation from CNN or ViT feature maps can be performed via soft weighting (inspired by differentiable rendering in 3DGS), enabling seamless integration into gradient-based training.
>
> - **Parameter efficiency:** A 2D Gaussian requires only five parameters to describe a spatial region—$\mu_x$, $\mu_y$, $\sigma_x$, $\sigma_y$, $\rho$—offering a compact yet expressive representation that avoids overburdening the generative decoder.
>
> In contrast, bounding boxes are limited to axis-aligned rectangles and exhibit hard, non-differentiable boundaries. Segmentation masks offer precise shapes but they are high-dimensional, discrete, and incompatible with differentiable optimization. Therefore, our use of 2D Gaussians is not merely inspired by 3D Gaussian Splatting (3DGS), but is also a well-motivated choice that satisfies the core demands of adaptive, efficient, and learnable tokenization.
>
> We will add more discussions in the revision, particularly in the introduction and method sections.
>
> ---
>
> **Q2: Missing ablation studies.**
>
> Actually, we have included a comprehensive analysis in Section C of the Appendix (starting at L42). This section presents both quantitative results and visualizations of the learned Gaussian parameters, which could help elucidate the role of each component in GPSToken. Specifically, the **"Refine."** module adjusts Gaussian distributions to better align with local textures and consistently improves performance across metrics. The **"Init."** module reallocates Gaussians from homogeneous to texture-rich regions, enabling finer semantic modeling without degrading reconstruction quality.
>
> To further justify our design and address the concern of this reviewer, we have conducted additional sensitivity studies on key hyperparameters, including entropy threshold $\lambda$, support factor $s$, minimum region size $s_{\min}$.
> Please refer to our response to Q2 of Reviewer EBwp, where we show that the model performance remains stable within reasonable ranges and our default settings achieve favorable performance.
> We will better highlight these results in the revision.
>
> ---
>
> **Q3: Why set $\sigma_x^{(i)}, \sigma_y^{(i)} = \frac{1}{6}w_i, \frac{1}{6}h_i$?**
>
> This is based on the **3σ rule** — a well-established empirical principle for normal distributions — which states that nearly **99.7%** of data points of a normal distribution lie within the range of $\mu \pm 3\sigma$. Based on this rule, each region is designed to have a width of approximately $6\sigma$ (spanning from $\mu - 3\sigma$ to $\mu + 3\sigma$), ensuring the inclusion of the vast majority of relevant data points.
> Notably, this initialization has minimal impact on the final outcome, as we subsequently refine the $\sigma$ values through transformer blocks in the encoder.
>
> ---
>
> **Q4: Performance on COCO and FFHQ.**
>
> As suggested, we further evaluate GPSToken on two additional real-world datasets: **COCO2017** and **FFHQ**, randomly sampling 5,000 images from each. As demonstrated in the following table, GPSToken consistently outperforms its competitors — MAETok (128 tokens) and VAVAE (256 tokens), representing the state of the art in 1D and 2D tokenization, respectively — across all metrics and datasets under the same token counts.
>
> | Dataset    | Token Count | Method         | PSNR ↑ | SSIM ↑ | LPIPS ↓ | rec. FID ↓ |
> |------------|-------------|----------------|--------|--------|---------|------------|
> | COCO2017   | 128         | MAETok         | 22.67  | 0.623  | 0.101   | 8.91       |
> |            |             | GPSToken-M128  | 23.47  | 0.657  | 0.083   | 4.72       |
> |            | 256         | VAVAE          | 25.01  | 0.736  | 0.052   | 6.01       |
> |            |             | GPSToken-L256  | 27.41  | 0.794  | 0.035   | 2.23       |
> | FFHQ       | 128         | MAETok         | 25.53  | 0.707  | 0.064   | 4.66       |
> |            |             | GPSToken-M128  | 26.35  | 0.745  | 0.050   | 3.72       |
> |            | 256         | VAVAE          | 28.06  | 0.808  | 0.027   | 1.95       |
> |            |             | GPSToken-L256  | 30.02  | 0.846  | 0.019   | 1.51       |
>
> Specifically, GPSToken-L256 achieves the best results on all three benchmarks: 30.02 PSNR / 0.846 SSIM on FFHQ, 28.81 PSNR / 0.809 SSIM on ImageNet (see Tab. 1 in the main paper), 27.41 PSNR / 0.794 SSIM on COCO2017.
> The observed performance trend -- FFHQ $>$ ImageNet $>$ COCO2017 -- aligns well with the inherent data complexity. The structured nature of human faces in FFHQ facilitates reconstruction, whereas ImageNet, with its 1,000 diverse object categories, presents a greater challenge. COCO2017, featuring complex scenes with multiple objects at varying scales and rich contextual interactions, poses the most difficult reconstruction task.
>
> The consistent superiority across datasets with different characteristics underscores the generalization capability of GPSToken.
>
> ---
>
> **Q5: More discussion on limitation.**
>
> Thanks for the constructive comments. We will include more discussions in the **Limitations** section based on this reviewer's suggestion.
>
> **Potential Failure Cases.**
> GPSToken adaptively allocates fewer tokens to simple regions and more tokens to complex regions for efficient representation without compromising reconstruction. One limitation of this strategy is that it may offer limited benefit for uniformly simple images and allocate dense tokens everywhere for images with high-frequency details or noise, reducing its efficiency. Our method is robust to occlusions because token allocation depends on the observed complexity of the visible region, but not predefined object structures. Changes in local content caused by occlusion naturally influence the token distribution based on the resulting complexity. We will show examples of these cases in the revision.
>
> **Token counts.**  GPSToken supports flexible token allocation during inference, allowing adaptation to varying computational budgets or fidelity requirements. As demonstrated in Fig. 6 of the main paper, reducing the total number of tokens preserves global structure while gradually sacrificing fine-grained local details. In contrast, increasing the token count will enhance the local detail reconstruction across the image.
>
> Crucially, this adaptability does not introduce significant spatial bias in token distribution. Our region partitioning strategy balances both spatial extent and local complexity, with a minimum region size constraint during initialization. This design ensures that even under low-token regimes, no region will be entirely omitted, thereby maintaining a spatially balanced representation.
>
> **Higher resolution.** GPSToken demonstrates favorable scalability to higher-resolution images. Empirically, we observe that reconstruction quality (measured by PSNR and SSIM) remains consistent when scaling both image resolution and token count proportionally. For instance, as shown in our response to Q4 of reviewer EBwp, reconstructing a $512 \times 512$ image using 512 tokens (or a $1024\times 1024$ image using 2048 tokens) achieves comparable fidelity to reconstructing a $256 \times 256$ image with 128 tokens. This result aligns with prior tokenization methods, suggesting that our approach preserves representational efficiency across resolutions.

---

> ### Author Response · Authors · 2025-08-04
>
> Dear Reviewer pWj1,
>
> Many thanks for your time in reviewing our paper and your constructive comments. We have submitted the point-to-point responses. We appreciate if you could let us know whether your concerns have been addressed, and we are happy to answer any further questions.
>
> Best regards,
>
> Authors of paper \#1542

---

### Official Review · Reviewer_EBwp · 2025-07-05

**Clarity:** 3
**Significance:** 3
**Originality:** 3
**Rating:** 4
**Confidence:** 2

**Summary:**

The paper proposes GPSToken, a way to break an image into variable-sized “Gaussian” tokens instead of a fixed grid. Each token stores its position/shape (as a 2-D Gaussian) and a texture feature. A differentiable renderer turns these tokens back into an image-like map so standard networks can use them. For image generation, the authors first predict a coarse layout of Gaussians and then fill in texture with a diffusion model, achieving better FID scores on ImageNet-256 than previous tokenization methods while converging faster.

**Questions:**

- The tokenizer alone has >120 M parameters, and splatting requires per-pixel Gaussian evaluation. Please report (i) encoder/decoder FLOPs, (ii) GPU memory during training and inference, and (iii) wall-clock throughput compared with VQGAN and TiTok at equal image size. Demonstrating competitive efficiency would strengthen the practical impact.

- Can GPSToken scale to 512×512 or 1024×1024 without exploding token counts? A small experiment or theoretical discussion on how σ scales with resolution—and whether the same model can tokenize non-natural images (e.g., medical, satellite)—would clarify robustness.

**Ethical Concerns:**

["NO or VERY MINOR ethics concerns only"]

**Limitations:**

Yes

**Quality:**

3

**Strengths And Weaknesses:**

**Paper Strengths**
- Model is large (≈128 M tokenizer + 675 M generator) and still needs 64–256 tokens, so actual memory/flops vs standard VQ/VAE remain unclear.
- Many hyper-parameters (entropy threshold λ, support factor s, calibration algorithm) are given without sensitivity study.
- Strong quantitative gains over many recent 1-D/2-D tokenizers on ImageNet reconstruction & generation.
- Paper is well organized; code release promised.

**Weaknesses**
- Lacks wall-clock latency / memory benchmarks and statistical significance (no error bars).
- Speed-ups are only theoretical; pooling overhead & two extra mat-muls may erode gains for small images—no profiling provide

---

> ### Author Rebuttal · Authors · 2025-07-30
>
> We sincerely thank this reviewer for the constructive comments and suggestion. We hope our following point-to-point responses can address this reviewer's concerns.
>
> ---
>
> **Q1: Params, FLOPs, memory, latency and throughput.**
>
> In terms of model size, our GPSToken tokenizer contains approximately 128M parameters, which is comparable to GaussianToken (130M) and larger than VQVAE-f16 (89M). Meanwhile, TiTok-B64 has 205M parameters and FlexTok has 950M. Overall, GPSToken is highly competitive in model scale.
>
> As shown in the following table, GPSToken has moderate computational cost. While the FLOPs of its decoder are higher than those of VQVAE-f16 and TiTok-B64, its latency is competitive with GaussianToken and significantly better than FlexTok, which employs a heavy autoregressive decoder. In terms of memory, GPSToken uses less GPU memory than FlexTok and GaussianToken during inference, and less memory than VQVAE-f16 during training, exhibiting favorable memory efficiency in both phases.
>
> | Method | Params (M) | FLOPs (G) Encoder | FLOPs (G) Decoder | Memory (MB) Train | Memory (MB) Inference | Latency (ms) | Throughput (sample/s) |
> | - | - | - | - | - | - | - | - |
> | VQVAE-f16 | 89.6 | 556 | 1014 | 78222 | 2627 | 72$\pm$0.05 | 111.44 |
> | TiTok-B64 | 204.8 | 220 | 973 | 45892 | 2359 | 96$\pm$0.08 | 83.44 |
> | GPSToken-M128 | 127.8 | 383 | 2689 | 50793 | 2567 | 180$\pm$0.15 | 44.56 |
> | GaussianToken | 130.6 | 1706 | 2285 | 60781 | 5352 | 181$\pm$0.17 | 44.32 |
> | FlexTok | 949.7 | 283 | 7665 | - | 7275 | 2714$\pm$3.11 | 2.88 |
>
> *Table: Comparison of parameter size, computational cost, memory usage, latency, and throughput for generating $256\times256$ images on an A100 GPU. Batch size is 8 for inference and 16 for training. Latency is averaged over 20 runs with standard deviation reported. Throughput is measured in samples per second. Training memory for FlexTok is unavailable due to lack of released training code.*
>
> While GPSToken ranks mid-tier in standalone efficiency, it is worth mentioning that tokenizers typically play a minor role in the overall latency of generation pipelines. The diffusion generator dominates the inference time; hence, efficiency is not our primary design goal. For instance, generating eight $256\times256$ images takes the diffusion model about 60 seconds, while decoding the latent with GPSToken requires only 0.15 seconds, which is negligible compared to the total latency.
>
> We hope the above analysis clarifies the efficiency profile of GPSToken and demonstrates its practical viability in real-world pipelines.
>
> ---
>
> **Q2: Hyper-parameter experiments.**
>
> The experimental results on $\lambda$, $s$, and $s_{\min}$ (hyper-parameter in calibration algorithm) are as follows.
>
> | Hyper-parameter | PSNR | SSIM | LPIPS | rec. FID | FID |
> | - | - | - | - | - | - |
> | $\lambda$ = 0 | 23.52 | 0.638 | 0.110 | 1.02 | 2.59 |
> | $\lambda$ = 5 | 24.06 | 0.652 | 0.083 | 0.68 | 2.23 |
> | $s$ = 1 | 17.55 | 0.4388 | 0.3435 | 160 | 165 |
> | $s$ = 3 | 24.07 | 0.657 | 0.080 | 0.66 | 2.18 |
> | $s$ = 7 | 24.06 | 0.658 | 0.080 | 0.66 | 2.16 |
> | $s_{min}$ = 8 | 24.05 | 0.656 | 0.080 | 0.66 | 2.18 |
> | $s_{min}$ = 16 | 24.03 | 0.657 | 0.081 | 0.66 | 2.19 |
> | ours($\lambda=2.5, s=5, s_{min}=4$) | 24.06 | 0.657 | 0.080 | 0.65 | 2.18 |
>
> **Entropy threshold $\lambda$**: As stated in Eq. 4 of the main paper, $\lambda$ balances region size and complexity.
> A larger $\lambda=5$  encourages Gaussians to concentrate on complex regions, leading to minor performance degradation.
> In contrast, setting $\lambda = 0$ allocates Gaussians solely based on region size, resulting in a uniform spatial distribution.
> This causes a significant drop in performance: LPIPS increases from 0.08 to 0.11, and rec.FID rises from 0.65 to 1.02.
> We set $\lambda=5$ to 2.5 based on experimental experience.
>
> **Support factor $s$**: As stated in Eq. 2 of the main paper, $s$ controls the effective rendering support of each Gaussian.
> Performance degrades significantly when $s = 1$, but stabilizes for $s \geq 3$. This aligns with the $3\sigma$ rule --- 99.7\% of the mass of a 2D Gaussian lies within three standard deviations from the mean. To ensure full coverage, we set $s = 5$ in all experiments.
>
> **Minimal region size $s_{min}$**: We set $s_{min}$ in the calibration algorithm to match its value in the initialization algorithm, where $s_{min}$ determines the minimum width or height of each region. We observe that increasing $s_{min}$ from 4 to 16 results in negligible performance degradation. This is expected because, with 128 tokens representing a 256$\times$256 image, the average spatial extent per token is approximately 22$\times$22 pixels. Consequently, most of the segmented regions naturally have a width or height greater than or equal to 16, making the choice of $s_{min}$ within this range largely inconsequential for the final tokenization.
>
> ---
>
> **Q3: Speed-ups for two-stage generator.**
>
> Existing methods that use pooling layers for speed-up often sacrifice performance (e.g., DiT shows FID degradation from 19.47 to 43.01 with larger pooling strides). In contrast, our GPSToken delivers practical speed-ups in training.
>
> As shown in the following table, compared to the baseline (SiT-XL/2), our method achieves better FID (9.57 vs. 19.07 at 500K iterations) while reducing training time by 42% (128h vs. 219h) and boosting throughput by 73% (1.09 vs. 0.63 iters/s) with a reduced memory usage. These results confirm tangible speed-ups of our two-stage generator.
>
> | Method | Metric | 500K | 1000K | T-Mem | T-Thpt |
> |-|- |- |- |- |- |
> | Baseline | FID | 19.07 | 14.50 | 63684 | 0.63 |
> | | Time (h) | 219 | 439 | | |
> | Ours | FID | 9.57 | 7.61 | 41498 | 1.09 |
> | | Time (h) | 128 | 256 | | |
>
> *Notes: T-Mem: Training Memory (MB), T-Thpt: Training Throughput (iters/s)*
>
> ---
>
> **Q4: Higher resolution.**
>
> Thanks for the insightful question on the scalability of GPSToken to higher resolutions.
>
> Similar to prior work, GPSToken requires token count to scale linearly with pixel count for consistent reconstruction quality. As shown in the table below, 128 tokens for 256×256 images achieve performance (e.g., LPIPS: 0.080, FID: 2.18) comparable to 512 tokens for 512×512 or 2048 for 1024×1024. However, using only 128 tokens for 512×512 images sharply degrades performance (LPIPS: 0.274, FID: 21.0), indicating severe loss of detail.
>
> | Resolution       | Token Count | PSNR   | SSIM   | LPIPS | rFID  | FID   |
> |-|-|-|-|-|-|-|
> | $256 \times 256$ | 128         | 24.06  | 0.657  | 0.080 | 0.65  | 2.18  |
> | $512 \times 512$ | 128         | 21.92  | 0.586  | 0.274 | 18.13 | 21.01 |
> | $512 \times 512$ | 512         | 26.44  | 0.754  | 0.084 | 0.61  | 2.04  |
> | $1024 \times 1024$ | 512       | 23.47 | 0.729 | 0.257  | 7.52  | 9.54  |
> | $1024 \times 1024$ | 2048      | 27.16    | 0.848 | 0.093    | 0.88  | 2.32  |
>
> This is visually evident in Fig. 6 of the main paper (32 tokens at 256×256), where coarse structures remain but fine textures blur due to insufficient token density—similar to using 128 tokens for 512×512, supporting our observation.
>
> Notably, PSNR and SSIM improve at higher resolutions with linearly scaled tokens. This occurs because content within each token’s receptive field becomes smoother and less textured, easing reconstruction and inflating pixel-wise and structural similarity scores.
>
> Regarding the parameter $\sigma$, which controls the spatial extent of each GPS-token, we hypothesize that it should scale with the average area covered per token. Specifically, the average area per token is proportional to $H \times W / N$, where $H \times W$ is the image resolution and $N$ is the number of tokens. To maintain consistent local modeling capacity, $\sigma$ should scale with the square root of this area:
> $$
> \sigma \propto \sqrt{\frac{H \times W}{N}}.
> $$
> Thus, when resolution increases but $N$ remains fixed, $\sigma$ will increase to cover larger regions, resulting in over-smoothing and loss of detail. When both resolution and $N$ scale proportionally, $\sigma$ can remain unchanged, preserving local fidelity.
>
> ---
>
> **Q5: Non-natural images.**
>
> To evaluate the robustness and generalization of our GPSToken to non-natural images, we conducted experiments on public datasets: **STARE** (retinal fundus images in medical imaging) and **WHU_RS19** (satellite remote sensing images). We compare against VAVAE (256 tokens, a leading 2D tokenizer) and MAETok (128 tokens, a top-performing 1D tokenizer).
>
> In the table below, we report PSNR, SSIM, and LPIPS scores, which are more suitable for evaluating reconstruction quality on non-photorealistic images compared to metrics like FID.
>
> | Dataset        | Token Count | Method           | PSNR ↑ | SSIM ↑ | LPIPS ↓ |
> |-|-|-|-|-|-|
> | STARE          | 128         | MAETok           | 32.98  | 0.818  | 0.051  |
> |                |             | GPSToken-M128    | 34.75  | 0.868  | 0.036  |
> |                | 256         | VAVAE            | 36.32  | 0.896  | 0.019  |
> |                |             | GPSToken-L256    | 37.60  | 0.915  | 0.014  |
> | WHU\_RS19      | 128         | MAETok           | 21.73  | 0.506  | 0.195  |
> |                |             | GPSToken-M128    | 23.20  | 0.560  | 0.127  |
> |                | 256         | VAVAE            | 23.57  | 0.619  | 0.142  |
> |                |             | GPSToken-L256    | 26.33  | 0.731  | 0.064  |
>
> GPSToken consistently outperforms competitors at comparable token counts, e.g., achieving higher PSNR (26.33 vs. 23.57) than VAVAE on WHU_RS19. Due to space limits, visual comparisons are deferred to the revision. GPSToken-L256 also better reconstructs fine details (e.g., capillaries, trees, cars) while preserving large-scale structures.
>
> These findings suggest that GPSToken is not only effective on natural images, but also generalizes well to diverse image domains, making it a robust and versatile tokenizer for various vision tasks.

---

> ### Author Response · Authors · 2025-08-04
>
> Dear Reviewer EBwp,
>
> Many thanks for your time in reviewing our paper and your constructive comments. We have submitted the point-to-point responses. We appreciate if you could let us know whether your concerns have been addressed, and we are happy to answer any further questions.
>
> Best regards,
>
> Authors of paper \#1542

---

### Author Response · Authors · 2025-08-07

Dear Reviewers of paper \#1542,

Many thanks for your time and engagement in reviewing our manuscript. We have uploaded our responses to your valuable comments. Since the deadline of the reviewer-author discussion period is coming soon, we deeply appreciate if you could let us know whether your concerns have been addressed. We are also happy to answer any further questions.

Best regards,

Authors of paper \#1542

---

### Note · Authors · 2025-08-12

Dear Reviewers and Area Chairs,

We sincerely thank all reviewers for their valuable feedback and engagement during the rebuttal phase. We are encouraged by the reviewers' positive comments and recognition on our paper's novelty (`xDKG`, `pm2Q`), value (`pWj1`), effectiveness (`EBwp`, `pWj1`, `xDKG`, `pm2Q`), presentation (`EBwp`, `pWj1`, `xDKG`, `pm2Q`), and open-sourcing (`EBwp`).

In summary, we proposed a novel Gaussian Parameterized Spatially-adaptive Tokenization (GPSToken) framework to achieve non-uniform image tokenization. Our method breaks the limit of rigid and uniform 2D/1D grids of existing tokenization approaches and demonstrates substantial improvements in both reconstruction and generation tasks. We hope that our work can inspire more research in non-uniform and region-adaptive image tokenization, as well as the associated image generation techniques.

Best regards,

Authors of Paper \#1542

---

### Decision · Program_Chairs · 2025-09-17

**Decision:**

Accept (poster)

**Comment:**

This paper tackles spatially-adaptive image tokenization with a Gaussian parameterized approach. In contrast to fixed spatial correspondence, tokens are initialized with an iterative splitting algorithm to represent regions of variable size. Empirical evidence shows its advantage in reconstruction quality, generation quality, and training convergence speed. The final version should reflect the suggestions from reviewers and the promises in the rebuttal.